# A Mountain-Induced Moist Baroclinic Wave Test Case for the Dynamical Cores of Atmospheric General Circulation Models

Owen K. Hughes and Christiane Jablonowski

Department of Climate and Space Sciences and Engineering, University of Michigan, Ann Arbor, USA

**Correspondence:** Owen K. Hughes (owhughes@umich.edu)

**Abstract.**

Idealized test cases for the dynamical cores of Atmospheric General Circulation Models are informative tools to assess numerical designs' accuracy and investigate the general characteristics of atmospheric motions. A new test case is introduced that is built upon a baroclinically unstable base state with an added orographic barrier. The topography is analytically prescribed and acts as a trigger of both baroclinic Rossby waves and inertia-gravity waves on a rotating, regular-size planet. Both dry and idealized moist configurations are suggested. The latter utilizes the Kessler warm-rain precipitation scheme. The test case enhances the complexity of the existing test suite hierarchy and focuses on the impacts of two midlatitudinal mountain ridges on the circulation. Selected simulation examples from four dynamical cores are shown. These are the Spectral Element and Finite Volume dynamical cores, which are part of NCAR's Community Earth System Model (CESM) versions 2.1.3 and 2.2, and the Cubed-Sphere Finite Volume dynamical cores, which is new to CESM version 2.2. In addition, the Model for Prediction Across Scales (MPAS) is tested. The overall flow patterns agree well in the four dynamical cores, but the details can vary greatly. The examples highlight the broad palette of use cases for the test case and reveal physics-dynamics coupling issues.

## 1 Introduction

An important component of an Atmospheric General Circulation Model (AGCM) is the dynamical core, which solves the fluid flow equations on a computational grid. The dynamical core thereby captures the resolved scales of the flow, defines the accuracy of the horizontal, vertical, and temporal numerical discretizations, determines the dissipation characteristics of the flow, and also selects the treatment of topography via the choice of the vertical coordinate. Testing the accuracy of a dynamical core is a paramount development step for weather and climate models. This is typically facilitated by performing dynamical core integrations of idealized test cases. These test cases have lower complexity than realistic weather forecasts or climate simulations and, for example, use only dry dynamical core configurations, dry or moist model setups with simplified physical processes or simplified lower-boundary conditions, and/or idealized initial conditions. This provides a controlled environment that captures selected atmospheric motions of interest. Such idealized model configurations serve two purposes. First, they allow assessments of the numerical schemes and serve as a standardized testing framework for model intercomparisons, thereby guiding developers' design and tuning decisions. Second, idealized test cases are also used as atmospheric dynamics tools to

understand physical phenomena, such as the dependence of orographic gravity waves on the Froude number, or to assess the impacts of mountains on midlatitudinal dynamics, precipitation, or the general circulation of the atmosphere. Our proposed test case serves both purposes. The goal of this paper is to introduce a new test technique for the dynamical cores of atmospheric General Circulation Models. The novel approach is that we combine an existing baroclinic instability test case with idealized topographic barriers, which has been a missing link in the existing test case hierarchy. Selected examples are then used to illustrate possible application areas.

The suite of test cases for dynamical core and idealized climate model validations spans a hierarchy of complexities. Test cases have, for example, been developed for the simpler shallow water equations (Williamson et al., 1992; Galewsky et al., 2004; Shamir et al., 2019), which serve as a 2D horizontal testbed for atmospheric motions. In addition, the hierarchy includes test cases for dry 3D dynamical cores (Held and Suarez, 1994; Jablonowski and Williamson, 2006; Wedi and Smolarkiewicz, 2009; Lauritzen et al., 2010; Kent et al., 2014a; Ullrich et al., 2014; Shamir and Paldor, 2016), idealized moist 3D dynamical cores (Thatcher and Jablonowski, 2016; Klemp et al., 2015), and aqua-planet models (Neale and Hoskins, 2000; Lee et al., 2008). Aqua-planet models use a full-complexity physical parameterization suite but a simplified lower boundary condition. The latter is either built upon a flat, ocean-covered earth with analytically prescribed sea surface temperatures as in Neale and Hoskins (2000) or utilizes a slab ocean configuration with a constant mixed-layer depth as in Lee et al. (2008) or Kang et al. (2008).

One dynamical core design aspect that can be studied at various levels of complexity is the treatment of topography and the vertical coordinate. Often, the inclusion of topography in a dynamical core is first tested via simpler equation sets that, for example, utilize a hydrostatic, Boussinesq, or anelastic approximation and set the Coriolis parameter to zero. Typically, 2D Cartesian x-z configurations with smoothly varying (e.g., bell-shaped) mountain profiles and idealized initial conditions with a constant background stratification and zonal flow are used. Examples are the 2D nonrotating test configurations by Klemp and Lilly (1978), Durran and Klemp (1983), Satomura et al. (2003) and Kurowski et al. (2013) that were designed for dry and moist orographic flows. Alternatively, Schär et al. (2002) and Guerra and Ullrich (2021) used dry, nonrotating, orographic gravity wave tests to assess their 2D x-z non-hydrostatic model designs. A portfolio of 2D hydrostatic and non-hydrostatic gravity waves, as well as inertia-gravity waves with rotation on a fixed $f-$plane, were assessed in Dudhia (1993) and Ullrich and Jablonowski (2012a). In addition, 3D Cartesian nonrotating mountain waves were analyzed in, e.g., Smolarkiewicz and Rotunno (1989) and Schär and Durran (1997). For such idealized test scenarios, linear as well as nonlinear analytic steady-state gravity wave solutions can be computed as shown in Smith (1980) and Guerra and Ullrich (2021), respectively.

However, dynamical core test cases for orographic flows on the sphere are less abundant in the literature. In general, three aspects are discussed. The first aspect addresses the accuracy of the vertical, often orography-following, coordinate and is sometimes called the "acid test". This assesses whether a resting nonrotating atmosphere in hydrostatic equilibrium stays motionless in the presence of topography as, e.g., assessed in Lin (1997), Qian et al. (1998), or Zängl (2012). The second test principle mimics the Cartesian gravity wave configurations mentioned above. Idealized ridge mountains or mountains with circular shapes are then embedded in idealized flows with a solid body rotation and constant stratification on a nonrotating planet with either a full-size or reduced-size radius. Such configurations were suggested in Tomita and Satoh (2004) (case

3), Ullrich et al. (2012) (case 2) and Klemp et al. (2015). In particular, the Ullrich et al. (2012) test variant was specifically developed for the "Dynamical Core Model Intercomparison Project" (DCMIP), which conducts regular international dynamical core assessments (see also Jablonowski et al. (2008), Ullrich et al. (2016, 2017) and Zarzycki et al. (2019)). The third test principle uses a full-size earth with the earth's rotation and focuses on the representation of orographically-induced Rossby

wave trains instead of gravity waves. Such test configurations with bell-shaped mountains were described in Tomita and Satoh (2004) (case 5), Jablonowski et al. (2008) (case 5), and Ullrich and Jablonowski (2012b). These are built upon highly idealized initial conditions, such as isothermal states, a constant stratification, and solid body rotation. The induced 3D Rossby wave train thereby mimics the widely-used 2D shallow water "test case 5" as defined in Williamson et al. (1992). However, test cases for more complex, analytically-prescribed initial flows with topography have not been described yet for spherical geometries.

Our proposed test case helps fill this gap and, in particular, assesses the impact of mountains on baroclinic waves for both dry and idealized moist dynamical core configurations.

Previous work in spherical geometry highlighted the design and usefulness of baroclinic wave test cases for atmospheric flows without orographic obstacles (Polvani et al., 2004; Jablonowski and Williamson, 2006; Staniforth and White, 2011; Ullrich et al., 2014, 2016). The life cycle of baroclinic waves can differ significantly depending on the structure of the baro-

clinically unstable atmosphere from which they develop (Thorncroft et al., 1993). In the absence of analytical solutions, the evolution of a baroclinic wave is then typically computed over 10-20 days and intercompared to numerical solutions from other dynamical cores to gain insight into the flow characteristics. This sheds light on the diffusivity of the models and can even reveal dynamical core design flaws as, for example, demonstrated in Williamson et al. (2009). This can also be used to determine adequate vertical grid spacing for a given grid resolution, such as in Iga et al. (2007). Adding 2D mountains to such

test configurations is not necessarily straightforward since the initial steady-state background conditions are analytically balanced and zonally symmetric. These characteristics of the initial conditions get disrupted by 2D mountain shapes. Therefore, orographic effects on idealized baroclinic waves have only been assessed in 3D Cartesian model configurations so far. The initial conditions are easier to balance in Cartesian geometry and have, for example, been used to study baroclinic waves and their interaction with a ridge mountain in Menchaca and Durran (2017, 2018).

The proposed test extends the test case hierarchy and describes the evolution of baroclinic waves on the rotating full-size planet, which are triggered by idealized topography. The background flow field is based upon the ideas in Staniforth and White (2011) and Ullrich et al. (2014, 2016) who defined a family of steady-state initial conditions for baroclinic waves without topography in dry and moist environments. In particular, the moist steady-state from Ullrich et al. (2016) is mainly utilized here in conjunction with a Kessler warm rain scheme. The latter represents an idealized parameterization of moisture processes

without a cloud phase (Kessler, 1969; Klemp et al., 2015) and was also used during DCMIP in 2016 (Ullrich et al., 2016). The idealized precipitation triggered by the baroclinic wave then amplifies the wave in a highly nonlinear way. However, the moisture processes are optional, and both dry and moist dynamical core evaluations with topography are insightful use cases. In this paper, we chose to add two mountain ridges in the northern midlatitudes, which require adjustments of the initial state to recover the well-balanced background condition for baroclinic waves. A broad palette of topographic shapes, peak heights,

and locations is possible as long as the topographic profile has an analytic description. The latter informs the computation of

the well-balanced, albeit not perfectly balanced, initial state. The mountains then act as triggers for baroclinic waves. They thereby replace the overlaid initial wind or temperature perturbations that are typically used in the absence of a topographic trigger.

In summary, this work introduces a test case that combines idealized moisture physics, topographic forcing, mountain-enhanced precipitation, and the evolution of baroclinic waves on a rotating full-size planet. The paper has three goals. First, we introduce the design of the mountain-induced baroclinic wave test case. Second, selected examples from the Spectral Element (SE, Lauritzen et al. (2018)) dynamical core of the Community Earth System Model (CESM) are used to illustrate the characteristics of the test case and its orographically-induced flow. Third, snapshots of a brief model intercomparison are shown to gain insight into various dynamical core designs and the associated model spread. This intercomparison includes simulations with the Model for Prediction Across Scales (MPAS, Skamarock et al. (2012)), as well as the CESM Finite-Volume (FV, Lin (2004)) and CESM Cubed-Sphere Finite-Volume (FV3, Harris et al. (2021)) configurations. The latter two are part of the CESM version 2.2 release of the National Center for Atmospheric Research (NCAR). The test case is expected to help diagnose numerical artifacts resulting from the inclusion of topography in dynamical cores and reveal physics-dynamics coupling issues. In addition, the test enables general assessments of the atmospheric circulation driven by mountain-generated gravity and Rossby waves. It thereby serves as a new generic tool in the atmospheric dynamics toolbox.

This article is structured as follows. Section 2 lays out the specifications of the test case and justifies the chosen parameters. Section 3 introduces the dynamical cores which are used for a brief model intercomparison. Section 4 analyzes the important characteristics of the orographically-induced baroclinic wave via the SE model. Section 5 highlights selected dynamical core intercomparisons and briefly surveys a physics-dynamics coupling aspect revealed by this test case. The appendices provide technical specifications for all four dynamical cores assessed here to make the results reproducible.

## 2   Test Case Design

Previously designed 3D dynamical core test cases (Jablonowski et al., 2006; Ullrich et al., 2014, 2016) have demonstrated that baroclinic waves are an efficient tool for assessing the characteristics of dry and moist flow fields. These test cases have two key components: a steady-state background state that is designed to be baroclinically unstable and an added perturbation that triggers the formation of a baroclinic wave. Our test case is designed with a moist and dry variant. In moist runs, Kessler physics (Kessler, 1969; Klemp et al., 2015; Ullrich et al., 2016) is chosen as the precipitation mechanism. It is an idealized warm-rain scheme with three water species: dry mixing ratios of water vapor, liquid water, and rain water without ice. The Kessler physics package is explained in Appendix A. No other physical parameterizations are employed. This test setup thereby sheds light on the impact of the diabatic forcing from the precipitation on the evolution of the wave and the physics-dynamics coupling strategy. The dry, adiabatic variant of the test case is obtained by simply setting the initial humidity content to zero and avoiding the use of physical parameterizations.

The design of the test case is inspired by real-world phenomena and topographic shapes like the Andes or the Rocky Mountains. In nature, extreme precipitation can result from topographic forcing, such as the interaction between atmospheric

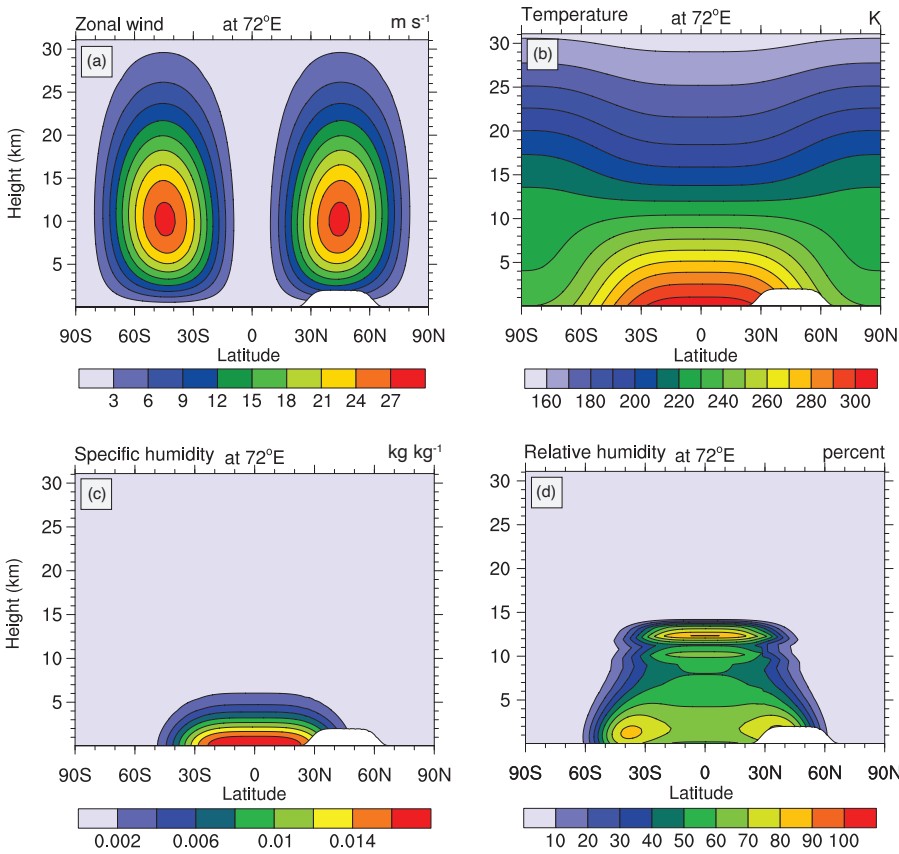

**Figure 1.** Latitude-height profiles at $72°$ E of the initial (a) zonal wind $u$, (b) temperature $T$, (c) specific humidity $q_v$, and (d) relative humidity. The idealized topographic profile is shown in white in the lower right of the plots.

rivers and mountains in the Pacific Northwest region of the United States. The test case is not designed to be complex enough to compare directly to real-world atmospheric rivers. However, the evolving precipitation bands that develop along with the topographically triggered baroclinic wave make our idealized test configuration a controlled setting for studying the effects of the dynamical core design on such high-intensity precipitation scenarios.

## 2.1 Properties of the initial background state

The atmospheric base state for the baroclinic wave without an overlaid perturbation is taken from Ullrich et al. (2014, 2016). They describe an analytic steady-state solution to the dry and moist 3D fluid flow equations on a rotating sphere without topography. Both shallow-atmosphere and deep-atmosphere dynamical core designs are accommodated. All base-state prognostic variables are zonally symmetric in the absence of topography. Because the base state is drawn from previous work, most

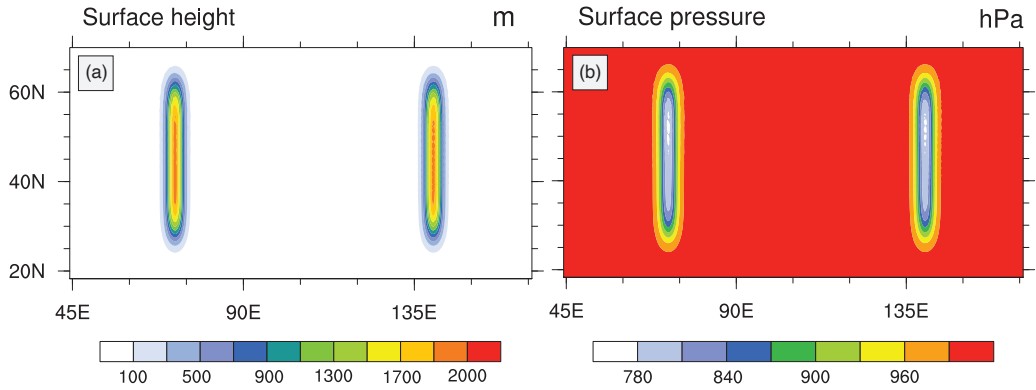

**Figure 2.** Latitude-longitude cross-sections of the (a) surface height $z_s$ and (b) initial surface pressure $p_s$.

functional forms for the prognostic variables are relegated to Appendix B. In particular, the appendix lists the equations for the temperature $T$, zonal wind $u$, meridional wind $v$, pressure $p$, density $\rho$, and specific humidity $q_v$ in Eqs. (B1)-(B6). The latitude-height (z) cross sections of the initial conditions along the longitude 72° E are shown in Fig. 1. This longitudinal location corresponds to the center position of the first mountain ridge, which is depicted by the white area (see also Sect. 2.2). As outlined in Ullrich et al. (2014, 2016), models with a pressure-based vertical coordinate can be initialized by using a numerical root-finding technique to solve for the height $z$ for any given pressure $p$ and then substituting this height value into the provided equations. Figure 1a shows that the zonal wind is characterized by westerly jets in the midlatitudes. Their vertical wind shear profiles support the growth of baroclinic instability waves. The temperature distribution $T$ (Fig. 1b) is in thermal wind balance with the zonal wind. The specific humidity $q_v$ (Fig. 1c) is chosen to resemble the zonal-mean distribution of water vapor. Above the artificial tropopause level $p_t = 150$ hPa, the $q$ field is set to zero as listed in Eq. (B6) and Table B1. We note that this setting deviates slightly from Ullrich et al. (2016), which specified a minimum stratospheric specific humidity value of $10^{-12}$ kg kg$^{-1}$ above 100 hPa. This change is irrelevant for the tropospheric baroclinic wave but prevents an initial supersaturation in the stratosphere. The moisture profile attains a maximum relative humidity of about 85% in the lower midlatitudes, as shown in Fig. 1d. This calculation makes use of Tetens' formula Eq. (B8) for the saturation condition as further explained in Appendix B.

### 2.2 Inclusion of topography: Surface height and surface pressure

The balanced background state is a steady-state solution in the absence of topography. The forcing by the added topography then triggers the baroclinic Rossby wave trains. The topographic profile and balanced surface pressure are shown in Fig. 2. These profiles utilize the mountain parameters and physical constants from Tables 1 and B1, and describe two non-overlapping ridges in the northern midlatitudes. The mountain shapes and peak heights impact the strength of the topographic forcing. They are chosen so that the baroclinic waves mature over the course of six days.

**Table 1.** Parameters for the test case. The degrees are specified in radians as needed by the equations.

| Variable Name | Variable Description | Value |
|---|---|---|
| $h_0$ | Peak mountain height | $2 \times 10^3$ m |
| $\phi_{1,2}$ | Latitude of the mountain peaks in radians | $\pi/4$ |
| $\lambda_{1,2}$ | Longitudes of the two mountain peaks in radians | $72\pi/180, 140\pi/180$ |
| $\overline{\lambda}$ | Nominal longitudinal width of the mountain in radians | $7\pi/180$ |
| $\overline{\phi}$ | Nominal latitudinal width of the mountain in radians | $40\pi/180$ |
| $d$ | Latitudinal scale parameter | $\frac{\overline{\phi}}{2}(-\log(0.1))^{-1/6}$ |
| $c$ | Longitudinal scale parameter | $\frac{\overline{\lambda}}{2}(-\log(0.1))^{-1/2}$ |

For the functional form of the topographic shape we define a modified longitude variable to make the description independent of the implemented longitudinal range of the model, such as $[0, 2\pi]$ or $[-\pi, \pi]$. Suppose that an AGCM parameterizes longitude over the interval $\lambda \in [\lambda_{\min}, \lambda_{\max}]$ and $\lambda_{\max} - \lambda_{\min} = 2\pi$. Then we define $d_n(\lambda) = (\lambda - \lambda_{\min}) - \lambda_n$, where $n \in \{1, 2\}$ indexes each mountain. The corresponding longitudinal center locations $\lambda_{1,2}$ are listed in Table 1. This leads to the modified longitude $l_n(\lambda) = \min(d_n(\lambda), 2\pi - d_n(\lambda))$ which ranges over the longitudinal distance $[-\pi, \pi]$ as measured from the longitudinal center point. The latitudes $\phi$ spans the interval $[-\pi/2, \pi/2]$. The mountain profile is then defined via the surface height

$$z_s(\lambda, \phi) = h_0 \sum_{n=1}^{2} \exp\left[ -\left( \left(\frac{\phi - \phi_n}{d}\right)^6 + \left(\frac{l_n(\lambda)}{c}\right)^2 \right) \right]. \tag{1}$$

Generally, any topographic profile is possible as long as it can be described by an analytical equation. The parameter $h_0$ represents the peak height of the topography. The functional form of each mountain, shown in Fig. 2a, is Gaussian in longitude. The exponent of the latitude term is increased to 6 from 2 to elongate the peak of the mountain meridionally. This elongation helps minimize any deviation of the maximum height of the discretized surface topography from the analytic maximum surface height $h_0$. The parameters $\phi_n$ and $\lambda_n$ represent the center latitude and longitude of the $n^{\text{th}}$ mountain, respectively. The parameter $\overline{\phi}$ specifies the distance along a line of constant longitude $\lambda = \lambda_n$ between the points where the surface topography is 10% of its maximum, that is, $z_s(\phi_n \pm \overline{\phi}/2, \lambda_n) = 0.1 \cdot h_0$. We treat this dimension as the nominal meridional extent of the mountain. The parameter $d$ transforms the specified $\overline{\phi}$ into the form required for the Gaussian-like functional form for the topography. Likewise, the parameter $\overline{\lambda}$ specifies the distance along a line of constant latitude $\phi = \phi_n$ such that $z_s(\phi_n, \lambda_n \pm \overline{\lambda}/2) = 0.1 \cdot h_0$, which is treated as the nominal zonal extent of the mountain. The parameter $c$ transforms the specified $\overline{\lambda}$ into the form required by the Gaussian functional form for the topography in the zonal direction.

The corresponding, balanced surface pressure can be calculated by evaluating the pressure profile from Eq. (B4) along the topographic profile:

$$p_s(\lambda, \phi) = p_0 \exp\left[ -\frac{g}{R_d} \left( \tau_{\text{int},1}(z_s(\lambda, \phi)) - \tau_{\text{int},2}(z_s(\lambda, \phi)) I_T(\phi) \right) \right]. \tag{2}$$

Figure 2b shows that the surface pressure varies from $p_0 = 1000$ hPa at sea level to about 773 hPa near the northern tip of the ridges.

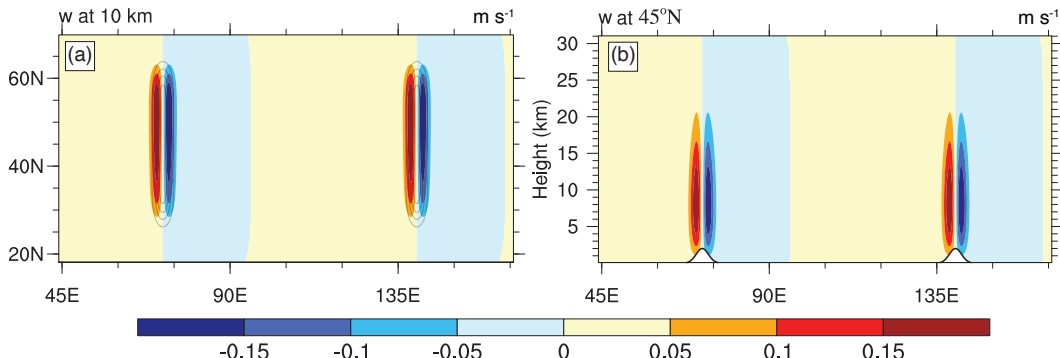

**Figure 3.** Cross-sections of the initial $w$ profile in Gal-Chen height coordinates for non-hydrostatic models. (a) Latitude-longitude cross-section at a height of 10 km and (b) Longitude-height cross-section at $45°$N. Topography is schematically shown by the grey contours in (a) and the white profile in (b). The computed extrema of $w$ are $\pm 0.209$ m s$^{-1}$.

### 2.3 Vertical velocity

In hydrostatic dynamical cores with a pressure-based vertical coordinate, the initial vertical pressure velocity $\omega$ does not need to be initialized. It will be computed diagnostically during the model integration. However, non-hydrostatic models must account

for the initial vertical velocity induced by the non-zero zonal wind along the sloping topographic boundary. A non-zero $w$ can be added so that the vector $[u, v, w]^\top$ runs parallel to vertically sloping model levels if a topography-following coordinate is used. This is achieved by setting $w = \boldsymbol{v}_h \cdot \nabla_{s^*} z$ where $\boldsymbol{v}_h$ symbolizes the horizontal wind vector at constant height $z$. The subscript $s^*$ denotes that the horizontal gradient must be computed along the transformed orography-following vertical coordinate, symbolically represented as an $s^*$ surface.

For our background state with zero meridional wind $v$, the vertical velocity for non-hydrostatic models can be expressed as

$$w = \frac{u}{a\cos\phi}\left(\frac{\partial z}{\partial \lambda}\right)_{s^*}, \tag{3}$$

which utilizes a spherical notation for the derivative in the zonal direction. The exact functional form for $w$ depends on the choice of $s^*$. However, for illustration purposes, a concrete example is displayed below. This closed form for $w$ is shown for the height-based orography-following Gal-Chen vertical coordinate $s^* = \overline{z}$ (Gal-Chen and Somerville, 1975; Kent et al., 2014a),

which is often used in non-hydrostatic models. The relationship between the geometric height $z$ and the transformed Gal-Chen coordinate $\overline{z} = z_{\text{top}}(z - z_s)(z_{\text{top}} - z_s)^{-1}$ is then given by

$$z = \overline{z} + (1 - \overline{z}/z_{\text{top}})z_s(\phi,\lambda) \quad \text{with} \quad \frac{\partial z}{\partial \lambda} = \frac{\partial z_s}{\partial \lambda}(1 - \overline{z}/z_{\text{top}}) \tag{4}$$

where $z_{\text{top}}$ symbolizes the constant height position of the model top and $\overline{z}$ is a constant along the sloping model levels. For the mountain profile shown in Eq. (1) the vertical velocity from Eq. (3) can then be expressed as

$$w(\lambda,\phi,z) = -\frac{u(\lambda,\phi,z)}{a\cos(\phi)}\left(2h_0\left(1 - \frac{z}{z_{\text{top}}}\right)\sum_{n=1}^{2}\left(\frac{\partial l_n}{\partial \lambda}\right)\left(\frac{l_n(\lambda)}{c^2}\right)\exp\left[-\left(\left(\frac{\phi - \phi_n}{d}\right)^6 + \left(\frac{l_n(\lambda)}{c}\right)^2\right)\right]\right) \tag{5}$$

where

$$\frac{\partial l_n}{\partial \lambda} = \begin{cases} 1 & \text{if } d_n(\lambda) < \pi \\ -1 & \text{if } d_n(\lambda) \geq \pi. \end{cases}$$

For the example of a Gal-Chen coordinate with $z_{\text{top}} = 31$ km, the magnitude and spatial structure of $w$ given by Eq. (5) is plotted in Fig. 3. The vertical velocity in Fig. 3 is shown for both mountains. Figure 3a shows a latitude-longitude profile at a constant geometric height of $z = 10$ km above mean sea level. Updrafts are observed on the upwind side west of the mountain peaks, and downdrafts are present on the downwind side east of the mountain peaks. Figure 3b shows a longitude-height cross-section at $45°$N. The Gal-Chen coordinate exhibits vertically-sloped model levels, which are present near the zonal jet maxima. This causes $w$ to achieve its peak magnitudes at the approximate height of the zonal jet. Other choices for transformed vertical coordinates are also popular, which let the terrain-following characteristics decay more rapidly from the surface, such as described in Klemp (2011) for MPAS. In this case, the maximum initial magnitudes of $w$ are expected to be located at a lower position in the atmosphere. If non-Gal-Chen coordinates are used, the expressions (4) and (5) need to be adjusted and might no longer have closed-form analytical descriptions.

The initial vertical velocities for the mountain profiles described in Eq. (1) are small. Therefore, we suggest that it is also acceptable to start these simulations with $w = 0$ m s$^{-1}$ if the the dynamical core and numerical scheme tolerate the initial imbalance. This is the case for MPAS. When comparing an MPAS $w = 0$ simulation with a simulation that used a numerically computed $w$ in MPAS, the evolutions of the baroclinic waves were almost indistinguishable (not shown). Therefore, the initialization of the non-zero $w$ profile can likely be omitted in most models for moderately steep mountain profiles with initial vertical velocities of order $10^{-1}$ m s$^{-1}$ or smaller. The initialization choice for $w$ must be documented when using the test case for non-hydrostatic configurations. For simplicity and to ease the comparison to other non-hydrostatic dynamical cores, all MPAS examples in this paper are shown for $w = 0$ m s$^{-1}$, which gets adjusted to the expected vertical updraft and downdraft patterns over one time step without triggering numerical noise. Our chosen other three dynamical cores are hydrostatic and compute the vertical velocity as a diagnostic quantity.

## 2.4   Design considerations

The moist test variant allows the examination of the interactions between subgrid-scale physical parameterizations and the dynamical evolution of baroclinic waves. In addition, the impact of topographic forcing on both dry and moist waves can be assessed. Several design considerations guide the choice of the parameters and functional forms of the initial conditions. As displayed in Fig. 2a the two mountain ridges are centered at $45°$ N, are separated by $68°$ in longitude, and have a peak height of 2000 m. The shape of each mountain is chosen to broadly resemble the mean height of real mountain ranges such as the Andes or the Rocky Mountains and to have a comparable nominal zonal extent of around 7 degrees in longitude. By design, but unlike the real mountain ranges on earth, a second ridge with an identical shape is placed to the east of the first mountain.

Although a single mountain is a sufficient perturbation to the steady state to trigger a baroclinic wave, adding a second mountain increases the utility of the test case in several ways. For notational convenience, we refer to the mountain centered

at $\lambda_1 = 72°$ E as Mountain 1 (M1) and the mountain centered at $\lambda_2 = 140°$ E as Mountain 2 (M2). The developing baroclinic wave downwind of M1 is called Wave 1; likewise, the wave downwind of M2 is called Wave 2. The evolution of Wave 1 is nearly identical to the wave downwind of Wave 2 until Wave 1 is forced over M2. The evolution of Wave 1 and Wave 2 can then be directly compared to determine the impact of the topographic lifting on the evolving Wave 1.

The longitudinal offset between the two mountains was chosen so that the band of large-scale precipitation along the leading frontal zone of Wave 1 has time to reach peak intensity and length before the precipitation band is forced over M2. This is shown in Figs. 4e-h that display the precipitation rates of the evolving waves at days 3, 4, 5, and 6, respectively. The topographic lifting of Wave 1 occurs before the wave breaking sets in slightly after that (around day 5.5-6). This destroys the coherent structure of the precipitation band. A full discussion of Fig. 4 and the flow characteristics is provided in Sect. 4.1.

In the dry configuration of this test case, the missing diabatic forcing from the precipitation slows down the growth rate of the waves, as shown later. This means wave breaking has not occurred yet when the dry variant of Wave 1 reaches M2. This allows high-resolution model runs to be used as a reference solution. Although mathematical convergence cannot be expected when moist physics is added, the model intercomparisons presented in Sect. 5 show that model statistics still allow insightful comparisons between the dynamical cores for up to six days.

## 3   Description of the Dynamical Cores

Before discussing the simulation results, we briefly introduce the four dynamical cores used in this study. Two of these dynamical cores are available as options in the CESM model (Danabasoglu et al., 2020) version 2.1.3 (CESM 2.2) and version 2.1.3 (CESM 2.1.3). A third dynamical core is new in CESM 2.2. In particular, the versions of these three dynamical cores in CESM 2.2 are embedded in the CESM atmospheric component, called the Community Atmosphere Model version 6 (CAM6). CAM6 includes the "Spectral Element" dynamical core SE (Taylor and Fournier, 2010; Lauritzen et al., 2018), "Finite Volume" model FV on a latitude-longitude grid (Lin, 2004), and the Finite-Volume Cubed-Sphere model FV3 from the Geophysical Fluid Dynamics Laboratory (GFDL) as described in Harris et al. (2021). In addition, we use the MPAS dynamical core (Skamarock et al., 2012), which is available as a development version in CAM6. However, for the comparisons here, the MPAS (version 7) simulations were performed with the stand-alone version of MPAS (Jacobsen et al., 2019). Due to the experimental nature of CESM2.2 at the beginning of this work, model simulations for SE and FV were performed in CESM version 2.1.3. All simulations are performed with 30 vertical levels. The hybrid pressure-based model level positions for SE, FV, and FV3 are listed in Reed and Jablonowski (2012) and are recommended to users of this test case. The model top lies near 2 hPa, corresponding to a model top height of around 35 km. MPAS uses a height-based vertical coordinate with a model top of about 31 km. This position corresponds to a top pressure of about 8 hPa. The relevant configuration details, as well as the namelist settings for all four dynamical cores, including the portfolio of the dynamics, physics, tracer, remapping, or acoustic time steps, are listed in Appendix C and the Tables C1-C4.

## 3.1 Spectral Element (SE)

The hydrostatic SE dynamical core in CESM is documented in Lauritzen et al. (2018) and was originally designed by Taylor et al. (1997) and Taylor and Fournier (2010). The spectral element method is also used in the Energy Exascale Earth System Model (E3SM) and supports non-hydrostatic extensions (Taylor et al., 2020). The spectral finite element method is formulated on an equiangular gnomonic cubed-sphere grid. Its horizontal discretization uses a mimetic A-grid with $4 \times 4$ continuous collocation points in each spectral element (the so-called "np4" configuration). This renders the numerical scheme fourth-order accurate in the horizontal direction. The numerical method exactly satisfies several differential identities that provide desirable conservation properties, such as the conservation of the dry air mass to machine precision. The continuous equations of motion also conserve a measure of moist total energy, which accounts for all prognostic water species (Taylor, 2011). The dry air mass is used to formulate the orography-following pressure-based vertical $\eta$ coordinate, which utilizes the Lorenz vertical staggering. The vertical discretization utilizes a floating Lagrangian coordinate similar to FV. All prognostic variables are then periodically remapped to their reference positions during a physics time step. Fourth-order hyperviscosity terms are added to the prognostic equations to prevent the accumulation of numerical grid-scale noise. The time stepping for the prognostic variables is done using a five-stage, nonlinear, third-order Runge-Kutta method.

Various physics-dynamics coupling strategies are available, controlled by a namelist parameter `se_ftype`. For `se_ftype=0`, the forcing due to physical parameterizations is distributed in equal increments (dribbled) and added to the prognostic variables during the integration of the sub-cycled dynamical core. For `se_ftype=1`, all physics adjustments are added as a lump adjustment after each physics time step. In the case of `se_ftype=2`, the forcing of mass quantities like moisture tracers are added via the `se_ftype=1` sudden adjustment strategy. In contrast, all other forcings for the, e.g., temperature or velocity, are dribbled in (`se_ftype=0`). This option is considered the "hybrid" option. We use `se_ftype=0` except in section 5.3 where the impact of the SE default `se_ftype=2` is demonstrated.

## 3.2 Finite-Volume (FV)

The FV dynamical core solves the hydrostatic primitive equations on a latitude-longitude grid using a flux-form semi-Lagrangian scheme and a floating Lagrangian vertical coordinate (Lin, 2004). It utilizes the Piecewise Parabolic Method (Colella and Woodward, 1984) to represent sub-grid flux distributions and is horizontally third-order accurate. The horizontal discretization uses a combined C–D grid staggering. The vertical treatment allows several Lagrangian dynamics steps to be taken before remapping the vertical levels to a reference grid. Nonlinear limiters within the finite volume method introduce implicit diffusion. Explicit fourth-order horizontal divergence damping is added to the model to prevent energy accumulation at the grid scale (Whitehead et al., 2011). Second-order horizontal divergence damping is applied in the top layers to decrease the impact of wave reflection from the model top. The dynamics are integrated on a shorter sub-cycled time step than the physics time step, and forcing due to microphysics is added to the prognostic variables as a lump adjustment after the physics time step (Neale et al., 2010).

## 3.3 Finite-Volume on a Cubed Sphere (FV3)

The FV3 dynamical core (Harris et al., 2021) was originally developed by NOAA's Geophysical Fluid Dynamics Laboratory and now serves as the fluid dynamics backbone of NOAA's "Unified Forecast System" (UFS) for weather prediction applications in the U.S. It shares many characteristics of the FV dynamical core. FV3 is a finite-volume model that can solve either the hydrostatic primitive equations or the non-hydrostatic shallow-atmosphere equations on an equiangular gnomonic cubed-sphere grid. Here, the hydrostatic version is chosen as implemented in CESM 2.2. Like FV, FV3 uses the Piecewise Parabolic Method on a C-D grid (Lin and Rood, 1997; Putman and Lin, 2007, 2009) and is horizontally third-order accurate. A floating Lagrangian vertical discretization is used. The cubed-sphere grid reduces the numerical difficulties posed by the pole point singularities in the FV latitude-longitude grid. To prevent the accumulation of noise at the grid scale, 6th-order horizontal divergence damping is activated. In addition, monotonicity constraints are used in the horizontal advection and vertical remapping algorithms, which implicitly adds viscosity to the model. As in FV, a second-order divergence damping mechanism is utilized as a sponge layer near the model top. In addition, Rayleigh friction is applied to the horizontal wind velocities in the sponge layer if the model level pressure is less than 7.5 hPa. In our L30 configuration, this only affects the topmost full model level. The maximum relaxation time is set to 10 days at the model top.

## 3.4 Model for Prediction Across Scales (MPAS)

MPAS (Skamarock et al., 2012) is a finite-volume model that solves the non-hydrostatic shallow-atmosphere equations. The horizontal discretization is built upon a centroidal Voronoi tesselation mesh with a staggered C-grid and is designed to use the mimetic so-called TRiSK discretization (Thuburn et al., 2009; Ringler et al., 2010). Horizontal advection is nominally third-to-fourth-order accurate. The vertical dimension is treated with a second-order finite volume method with a smoothed terrain-following geometric height coordinate as specified in Klemp (2011). Various smoothing options are available for the orography-following vertical coordinate called $\zeta$, which impact the accuracy of the numerical scheme. In our MPAS model simulations, we do not activate the smoothing and therefore convert to the Gal-Chen configuration shown in Eq. (4) with $\zeta = \overline{z}$. MPAS has several diffusion options to damp numerical noise, including a Smagorinsky-type eddy viscosity, fourth-order hyperdiffusion, and 3D divergence damping. Our MPAS model integrations are configured to use the Smagorinsky-type diffusion. A detailed discussion of MPAS' treatment of the physics tendencies can be found in Klemp et al. (2007).

## 4 Characteristics of the Test Case

For demonstration purposes, the evolution of the baroclinic wave is first discussed for the Spectral Element dynamical core. Any other dynamical core could have been picked. The simulations were run with nominal grid spacings of $1°$ (ne30), $0.5°$ (ne60), $0.25°$ (ne120), and $0.125°$ (ne240) and 30 vertical levels where the "neXXX" notation refers to the number of supporting spectral elements in the horizontal direction per cubed-sphere face. For example, the ne30 setting has $30 \times 30$ supporting elements per cubed-sphere face. The construction of the Gauss-Lobatto-Legendre points at which solutions are computed

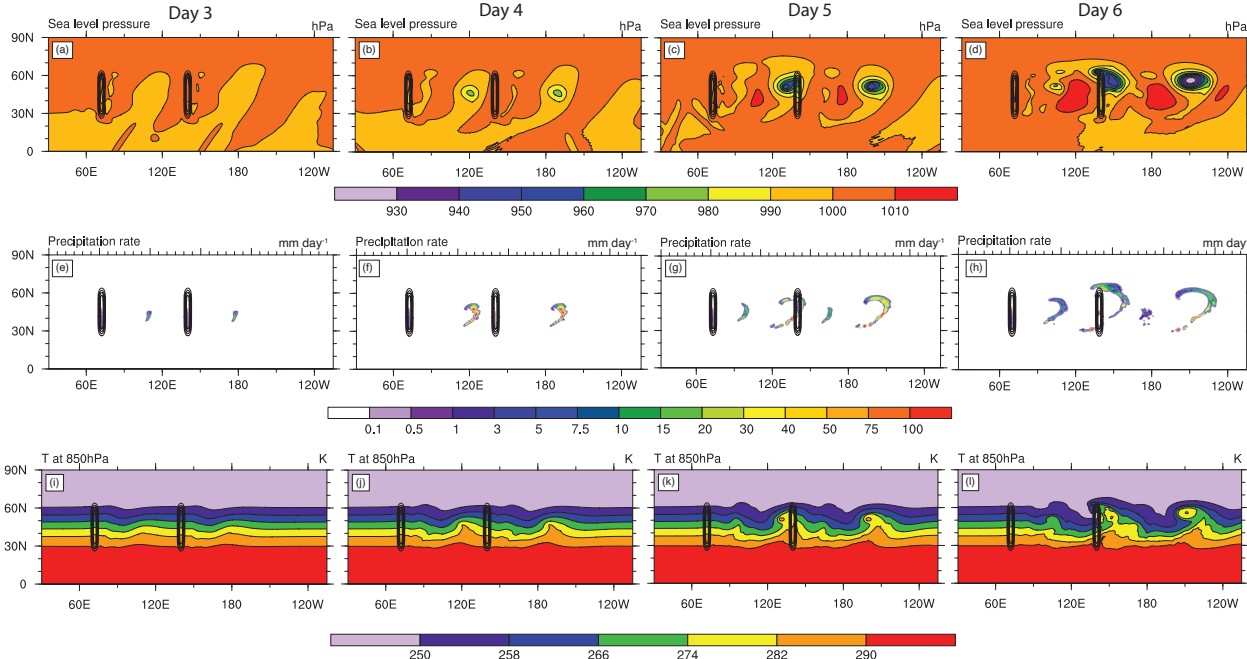

**Figure 4.** Latitude-longitude cross-sections of the baroclinic waves in the SE dynamical core on a $0.5°$ degree grid at days 3, 4, 5, and 6 (from left to right). Top row: mean sea level pressure, middle row: precipitation rate, bottom row: 850 hPa temperature. The contour lines indicate the location of the mountain ridges.

reduces nominal grid spacing by a factor of three (see Lauritzen et al. (2018) for further details). Therefore, the above grids have nominal geometric grid spacings of 100 km, 50km, 25 km, and 12.5 km, respectively. As mentioned before, our SE simulations used the `se_ftype=0` physics-dynamics strategy which deviates from the SE default `se_ftype=2`. The latter
default setting is explored in Sect. 5.3.

## 4.1 Baroclinic instability

Baroclinic instability is a crucial driver of weather systems in the midlatitudes. A systematic treatment of this phenomenon from the viewpoint of quasi-geostrophic theory can be found in, e.g., Holton (1992). Because the initial conditions in this test case are baroclinically unstable, each mountain triggers a synoptic-scale wave, which develops downwind of the topographic forcing.
Each wave exhibits characteristics of baroclinic waves in the real atmosphere. For example, strong temperature gradients develop ahead of the synoptic-scale low-pressure systems, which trigger strong precipitation bands along these frontal zones.

Figure 4 illustrates the time evolution of the mean sea level pressure (MSLP), precipitation rate, and 850 hPa temperature for the moist baroclinic wave at days 3, 4, 5, and 6. In particular, Figs. 4a-d show the intensifying low and high MSLP systems that develop behind both mountains. At day 4, the two developing low-pressure systems are nearly identical. At day 5, topographic
forcing begins to impact Wave 1 as Wave 1 is forced over M2. By day 6, topographic forcing has caused a significant deviation

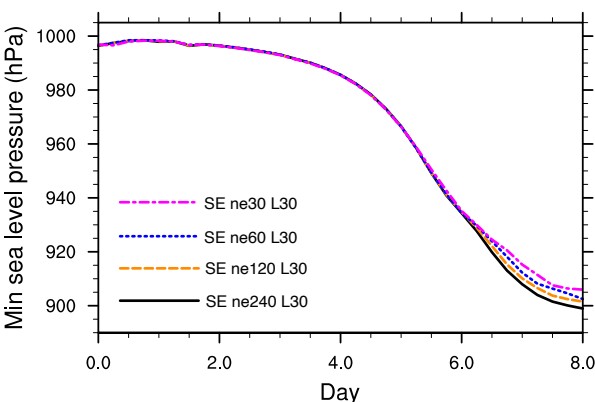

**Figure 5.** Time series of point-wise minimum MSLP over 8 days for a dry atmosphere over the SE resolution range ne30 - ne120 (100 - 12.5 km).

between the structure of the two waves. Figures 4e-h illustrate the development of the large-scale precipitation bands. These are fed by the high moisture transported from the tropical regions by the developing waves. At day 4, the bands begin forming to the east of the low-pressure system. At day 5, the precipitation band associated with Wave 1 is forced up over the mountain. By day 6, the topographic forcing has significantly disrupted the structure of the precipitation band associated with Wave 1 compared to the precipitation associated with Wave 2. Figures 4i-l show the evolution of the synoptic-scale temperature fronts at 850 hPa. Note that this 850 hPa position represents an interpolated level that uses extrapolation in the neighborhood of the mountain peaks. Nevertheless, we selected this low-lying level for the analysis as the wave signatures lose their sharpness with increasing altitude. These temperature fronts are driven by the transport of warm, moist, equatorial air into the midlatitudes, which in turn causes updrafts and the development of intense precipitation bands. The exponentially growing mode triggered by the addition of topography is well-resolved in horizontal grids with a $1°$ grid spacing. The agreement across resolutions breaks down in the moist case when wave breaking becomes dominant beyond day 6 (not shown).

In addition to the qualitative characteristics, several quantitative metrics are also assessed. Common quantities for assessing the development of baroclinic waves are the time evolution of the minimum MSLP and the Eddy Kinetic Energy (EKE) (Lorenz, 1955; Simmons and Hoskins, 1978; Pavan et al., 1999; Ullrich et al., 2014; Kurowski et al., 2015). Minimum MSLP measures the intensity of the most developed eddy and is calculated point-wise on an interpolated latitude-longitude grid. EKE measures the evolution of the kinetic wave energy relative to the background flow. It is computed in three steps. First, subtract the initial base state from the horizontal wind velocities $u$ and $v$ at each time slice. Second, calculate the point-wise kinetic energy of these eddy wind fields. Third, integrate the point-wise eddy kinetic energy over the entire volume of the atmosphere.

The calculation can be conducted in either height $z$ or pressure $p$ coordinates via

$$\text{EKE}(t) = \frac{1}{4\pi a^2} \int_{z_s}^{z_{top}} \int_A \frac{1}{2} \left[ \left( (u - \bar{u})^2 + (v - \bar{v})^2 \right) \rho \right] dA \, dz \tag{6}$$

$$= \frac{1}{4\pi a^2 g} \int_{p_{top}}^{p_s} \int_A \frac{1}{2} \left( (u - \bar{u})^2 + (v - \bar{v})^2 \right) dA \, dp \tag{7}$$

where $A$ denotes the area of a grid cell, and $\rho$ is the density of the air. The symbols $z_{top}$ and $p_{top}$ denote the height and pressure at the model top, respectively. The calculation only takes the horizontal velocities and their initial states $\bar{u}$ and $\bar{v}$ at each grid point into account and measures the EKE in units of J m$^{-2}$.

In the dry adiabatic configuration, point-wise convergence of EKE can be expected as the horizontal grid spacing is decreased. As was argued in Jablonowski and Williamson (2006) and Ullrich et al. (2014), this empirical point-wise convergence allows high-resolution model integrations to be used as reference solutions even when a closed-form solution for the evolution of the wave cannot be derived. Figure 5 shows a time series of the minimum MSLP in dry SE model integrations with decreasing horizontal grid spacing. The temporal progression of the minimum MSLP measured in the baroclinic wave converges

with increasing resolution. Therefore, the dry version of the test case can be used to benchmark the treatment of topography in the absence of moisture processes. When wave breaking sets in around and after day 6.5 in the dry configuration, the model solutions start to diverge due to the dominance of grid-scale turbulence and mixing. As an aside, the dry and moist baroclinic wave simulations without topography and an overlaid wind perturbation (Ullrich et al., 2014, 2016) start breaking between days 9 and 10. This shows that the presence of the large mountain ridges greatly accelerates the evolution of the waves while

using identical background states. In moist runs, the evolution of the wave is further accelerated and intensified by the diabatic heating from the precipitation, as shown in Fig. 6b and further discussed in the following subsection.

### 4.2    Impact of precipitation and orography

In the moist variant of the test case, the thermodynamic forcing caused by large-scale precipitation intensifies the development of the wave. Figure 6 shows the calculated minimum MSLP and EKE for the moist SE model integrations for decreasing

grid spacings. Unlike the dry case (Fig. 5), Fig. 6b illustrates that the minimum sea level pressure in the highest-resolution simulation diverges significantly from the lowest-resolution simulation once precipitation sets in between days 3 and 4. Figure 6a shows a time series of the integrated EKE. This demonstrates that the divergence of higher-resolution from lower-resolution model runs occurs over the whole wave structure, and the resolution dependence is not limited to the gridpoint at which MSLP is lowest. The EKE time series only illustrates the initial growth phase of the baroclinic wave. Saturation of the EKE values

occurs later, around day 10, with peak EKE values around $2 \times 10^5$ J m$^{-2}$ which compare well to the peak EKE values in Pavan et al. (1999).

     The strength of the eddy moisture flux convergence, which drives large-scale precipitation, is well-correlated with the diabatic heating in idealized studies of baroclinic modes (Pavan et al., 1999). This diabatic heating speeds up the growth rate of the wave. The correlation can be seen by examining the temperature anomaly, defined as the difference between the temperature at

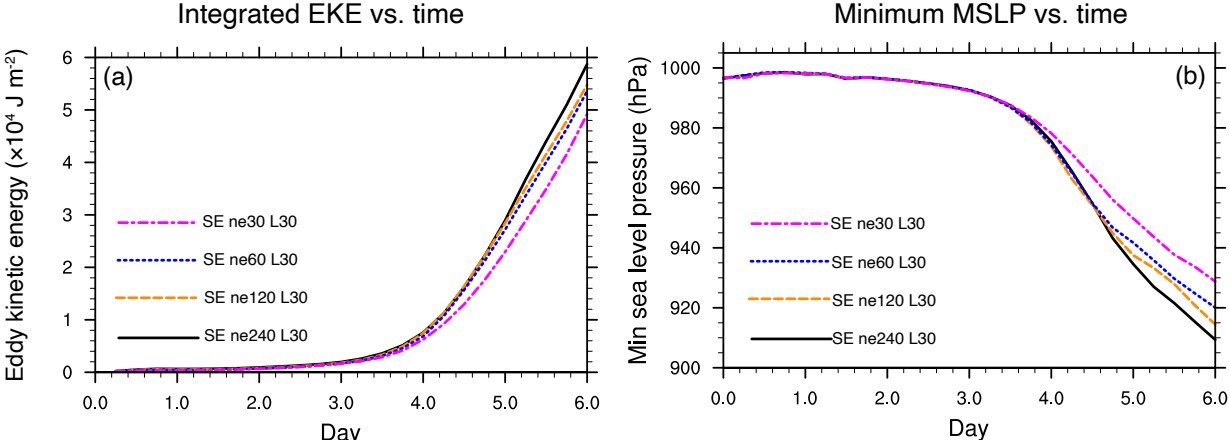

**Figure 6.** Time series of baroclinic wave summary statistics from the moist SE model at nominal $1°$ (ne30), $0.5°$ (ne60), $0.25°$ (ne120), $0.125°$ (ne240) grid spacings. (a) Eddy Kinetic Energy, and (b) point-wise minimum MSLP, which is a proxy for the amplification of the baroclinic wave.

a given time and the base state temperature. In addition, the vertical pressure velocity $\omega$ is a suitable proxy for the wave activity. The longitude-height cross sections of these fields are displayed at day 4 in Fig. 7 for both the dry and moist model integrations. Figure 7c shows that in the moist version, updrafts due to precipitation follow the progression of the temperature front. The updrafts are significantly larger in the moist case than in the dry case, as shown in Fig. 7c in the neighborhood of the frontal zone around $125°$ E. In addition, the $\omega$ patterns highlight the hydrostatic, mostly upward-propagating inertia-gravity wave

oscillations downwind of M2 near $140°$ E. These mountain wave patterns resemble the stationary hydrostatic inertia-gravity wave solutions from 2D x-z slice models on constant $f-$planes when tested with bell-shaped mountains (Dudhia, 1993; Ullrich and Jablonowski, 2012a). However, the spatial scale of the mountain-generated gravity waves in SE with full rotational effects is larger than that of the 2D models due to the different model setups. Figures 7a-b display the distribution of the temperature perturbation at day 4. The moist configurations in Fig. 7a shows that the diabatic forcing triggered by the precipitation com-

bined with the induced updrafts places the maximum positive temperature perturbation several kilometers into the atmosphere. The maximum positive temperature perturbation in the dry case (Fig. 7b) is located at the surface. The dependence of the wave intensification on the model resolution in Fig. 6 can be explained by noting that the maximum intensity of the extreme precipitation within the moisture bands increases as horizontal grid spacing decreases. We cannot expect point-wise convergence as grid spacing is decreased due to the nonlinearity of this forcing. However, it is reasonable to compare the statistics of the

precipitation between models in the absence of point-wise convergence. We demonstrate an example of such a comparison in Sect. 5.2.

The evolving circulation around the low-pressure systems induces moisture transport from the equatorial region when moisture is present. The circulation around the developing low-pressure systems creates bands of extreme precipitation to the east

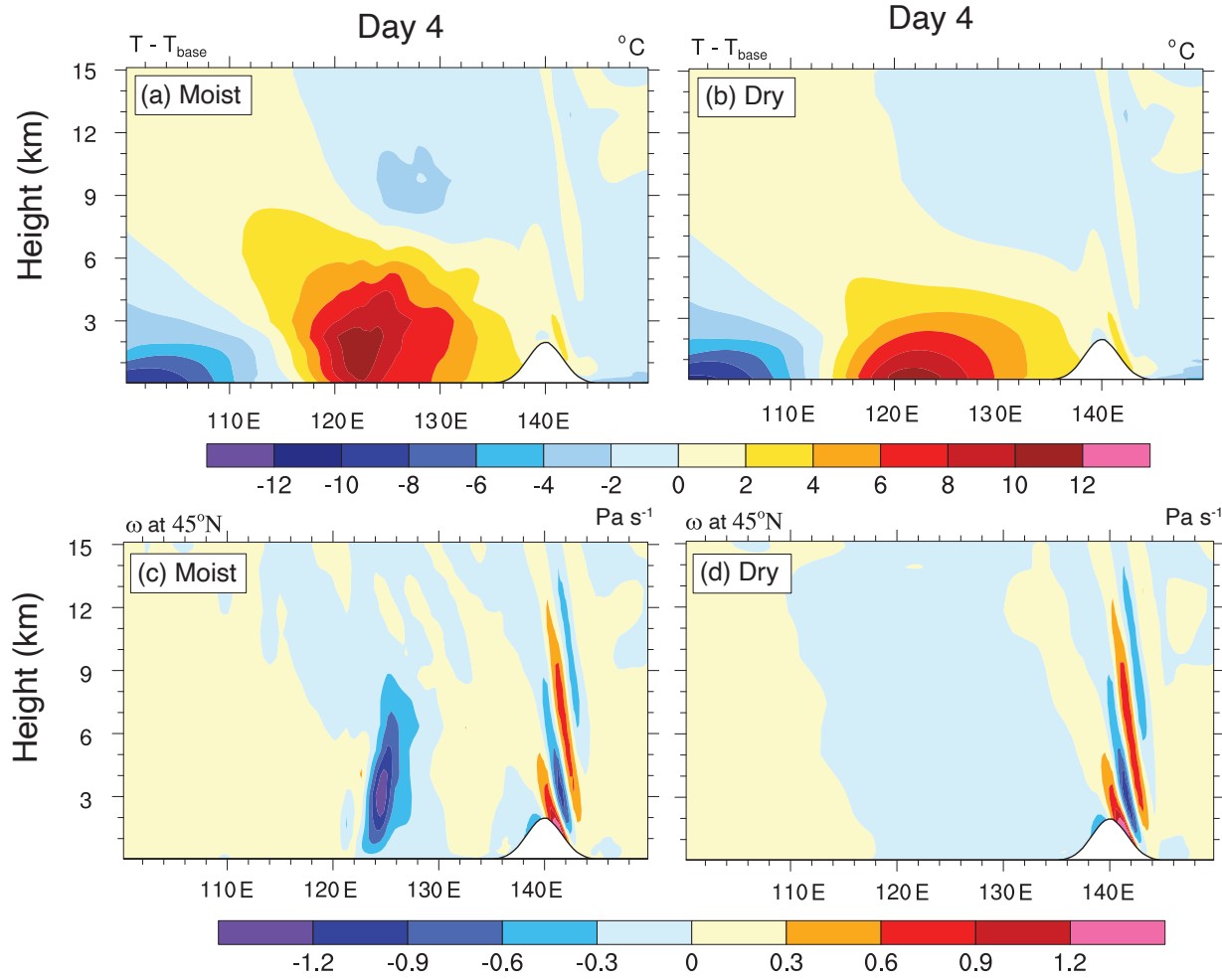

**Figure 7.** Longitude-height cross-sections at $45°$ N of the (top) temperature perturbation and (bottom) vertical pressure velocity $\omega$ for the (a,c) moist and (b,d) dry atmosphere. SE ne60 (50 km) model integrations at day 5 are shown.

of the low-pressure centers. The spatial extent of these precipitation bands reaches as far as $60°$ N. In addition, the bands reach
length scales of several thousand kilometers before wave breaking sets in (Fig. 4g). The leading band of Wave 1 is orographi-
cally lifted over M2 at day 5, which qualitatively mimics the impacts of the mountain ranges on atmospheric rivers along the
U.S. West Coast. Although the bands are narrow, the geographic distribution is well resolved even at the coarsest $1°$ horizontal
grid spacing. Because any sources of moisture are omitted in our simulations, water exits the atmosphere when precipitation
occurs. Surface fluxes of latent heat do not replenish it. Such a configuration with idealized surface fluxes represents a natural
extension of the test case complexity as described in Reed and Jablonowski (2012) and Thatcher and Jablonowski (2016) but
is not considered here.

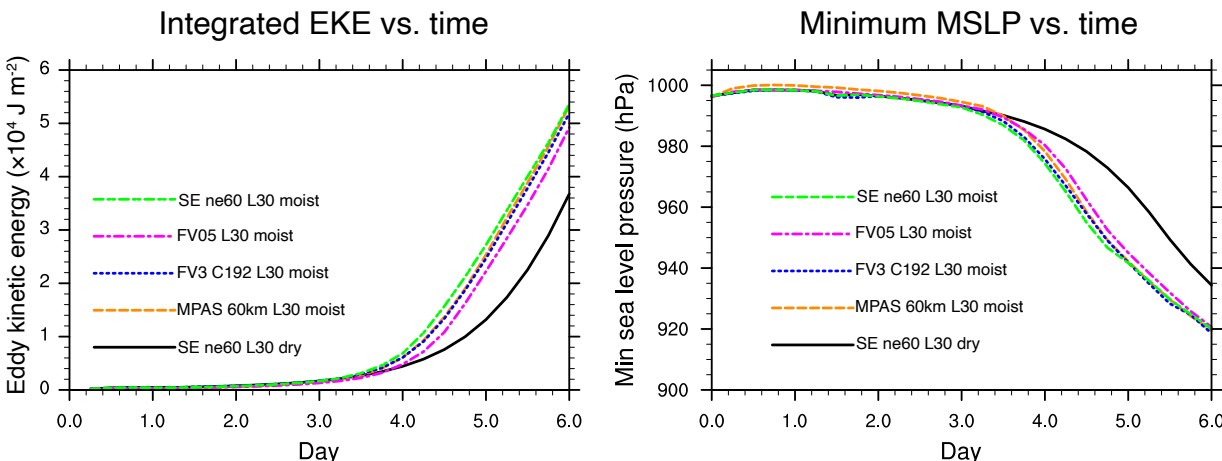

**Figure 8.** Time series of baroclinic wave summary statistics from moist model integrations with nominal $0.5°$ grid spacing in all four dycores. Evolution of (a) Eddy Kinetic Energy integrated over the entire volume of the model domain, and (b) the point-wise minimum MSLP. The time series for the corresponding dry SE $0.5°$ model integration is shown in black.

The presence of orography forces an upward motion of the precipitable water at day 5, thereby intensifying the precipitation rate as displayed in Fig. 4g. The comparison of the leading precipitation band triggered by M1 and the band triggered by M2 at day 6 (Fig. 4h) shows the reduction of the precipitation rate in Wave 1. This is caused by the orographic forcing of M2, which diminished the Wave 1 moisture pool compared to Wave 2. Furthermore, Fig. 4d illustrates that the interaction between the precipitation band and M2 slows down the intensification of the dominant Wave 1 low-pressure system.

## 5 Selected Dynamical Core Intercomparisons

Besides SE, we also tested the moist variant of the test case with FV, FV3, and MPAS to conduct a brief, non-exhaustive dynamical core intercomparison. Here, we provide selected snapshots of this intercomparison to highlight the capabilities of the test case. The simulations are conducted with 30 vertical levels and a nominal $0.5°$ grid spacing in all dynamical cores, which are labeled as "ne60" for SE, "FV05" for FV, "C192" for FV3, and "60km" for MPAS. The SE, FV3, and MPAS dynamical core analyses utilize model data on interpolated latitude-longitude grids with uniform $0.5° \times 0.5°$ horizontal grid spacings. The FV05 simulation uses the grid spacings $0.47° \times 0.625°$ for its latitude-longitude grid.

### 5.1 Baroclinic wave metrics

Quantitative metrics can be used to compare the strength of an evolving baroclinic wave across dynamical cores. Although Fig. 6 shows a significant dependence of the wave intensification to the horizontal grid spacing in the SE model, comparisons can be made across dynamical cores if the horizontal grid spacings are comparable. Figure 8 shows a time series of the evolution

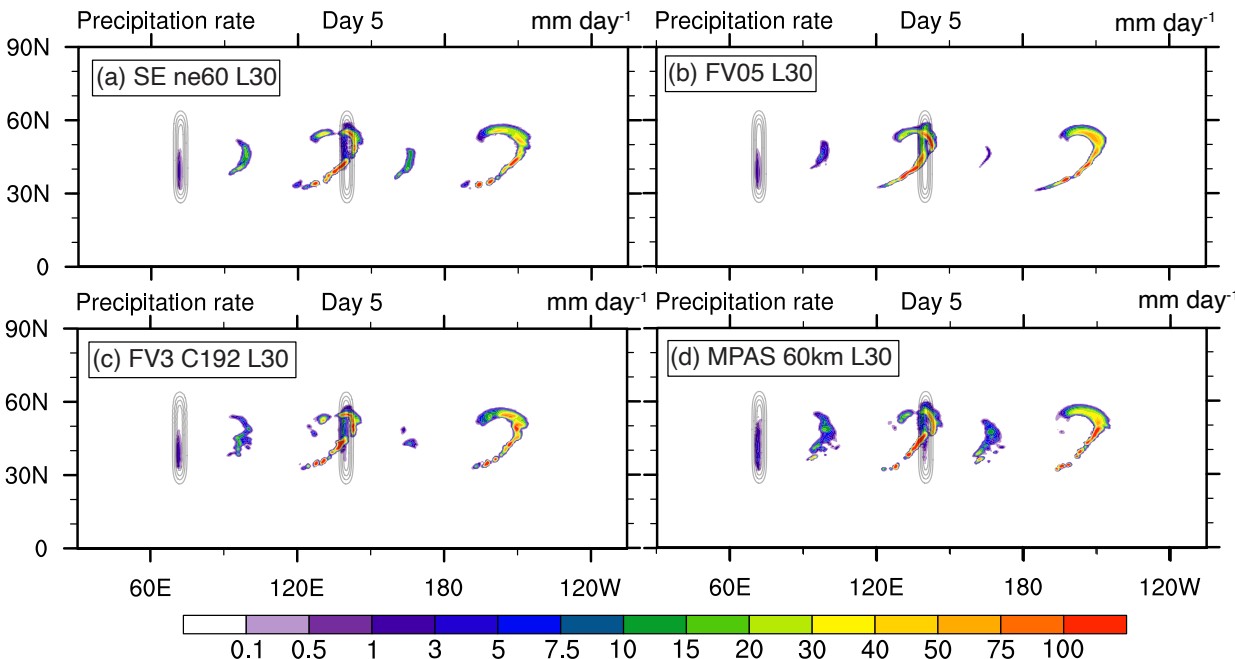

**Figure 9.** Intercomparison of the precipitation rates with nominal $0.5°$ grid spacings in (a) SE, (b) FV, (c) FV3, and (d) MPAS at day 5. The light contours mark the locations of the mountain ridges.

of EKE and minimum MSLP for the four moist dynamical cores over six days. In addition, the dry SE simulation is depicted to illustrate the differences between the moist and dry simulations. This shows the slower growth rate of the waves in the dry

configuration. The time evolution of both the EKE (Fig. 8a) and minimum MSLP (Fig. 8b) metrics are tightly clustered until intense precipitation develops along the frontal zones in the moist runs after day 3. With the onset of precipitation, the evolution in the FV dycore diverges from the others. The FV model integrations use fourth-order horizontal divergence damping. This is a more scale-selective dissipation process than the second-order horizontal divergence damping that is typically the default in FV. However, the slower amplification of the minimum MSLP in the FV dycore indicates that the model is still more diffusive

than the other dynamical cores. The evolution of the integrated EKE is less sensitive to isolated point-wise changes in the wave structure. The decreased EKE in the FV integration indicates that increased diffusion slows the rate of intensification across the entire wave pattern. After Wave 1 passes over M2 (at and after day 5) and, not taking FV into account for this discussion, the EKE spread between the dynamical cores increases, which is a consequence of the more and more dominant nonlinear effects. The minimum MSLP spread also increases at this point. However, this mostly happens after day 6 and is therefore less evident

in Fig. 8b.

Figure 9 shows an intercomparison of the precipitation bands at day 5 when Wave 1 is being orographically lifted. The most intense precipitation rate is observed on the upwind side of M2. However, the precipitation at the leading edge of Wave 1 on the downwind slope is still significantly more intense than the precipitation rate in the leading edge of Wave 2. The differences

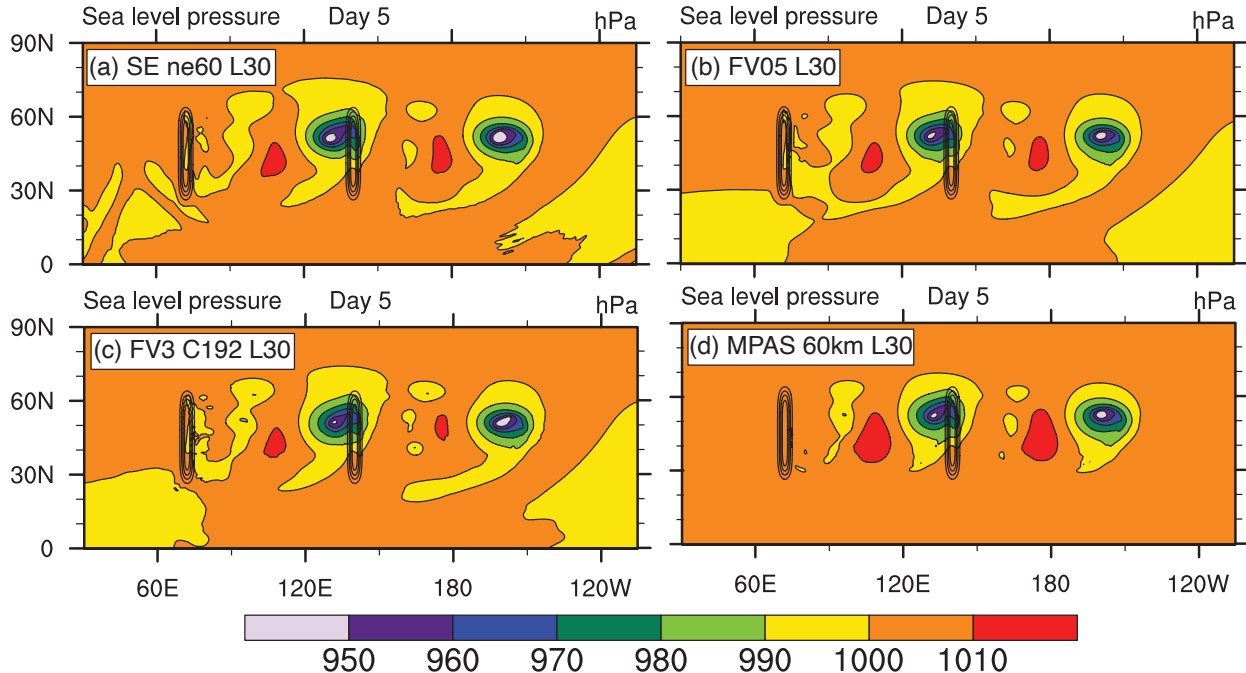

**Figure 10.** Intercomparison of latitude-longitude MSLP cross sections at day 5 from the moist (a) SE, (b) FV, (c) FV3, and (d) MPAS simulations with nominal 0.5° grid spacings. The oval-shaped contours mark the locations of the mountain ridges.

in the flow patterns are amplified by the highly nonlinear forcing provided by the combination of the topographically induced vertical motion and the diabatic forcing resulting from the increase in precipitation. We observe that the dynamical cores differ in their ability to keep the long precipitation bands together as coherent structures before wave-breaking processes break them up after day 5. For example, the precipitation bands in SE and MPAS in Fig. 9 already start developing small-scale but intense precipitation patches at day 5 that got separated from the main bands. These patches resemble so-called "grid-point storms", which are characterized by intense, truncation-scale storms with extreme updraft speeds and precipitation rates as analyzed in Williamson (2013). The coherent precipitation patches in FV and FV3 also break up due to wave breaking and stretching, but this happens slightly later. The reasons for these differences are complex, and an in-depth analysis is beyond the scope of this paper. However, the differences are likely caused by a combination of the following factors: insufficient resolution to represent the thin bands, the simplicity of the precipitation scheme, and the choice of the physics and dynamics time steps. These factors are tightly coupled to the differences in the diffusion characteristics and the associated so-called "effective resolutions" of the dynamical cores (see also Jablonowski and Williamson (2011) and Kent et al. (2014b, c)), and the physics-dynamics coupling strategies (see also Gross et al. (2016, 2018)). The physics-dynamics coupling aspect will be briefly highlighted for the SE dynamical core in Sect. 5.3, which sheds light on additional application areas for the test case.

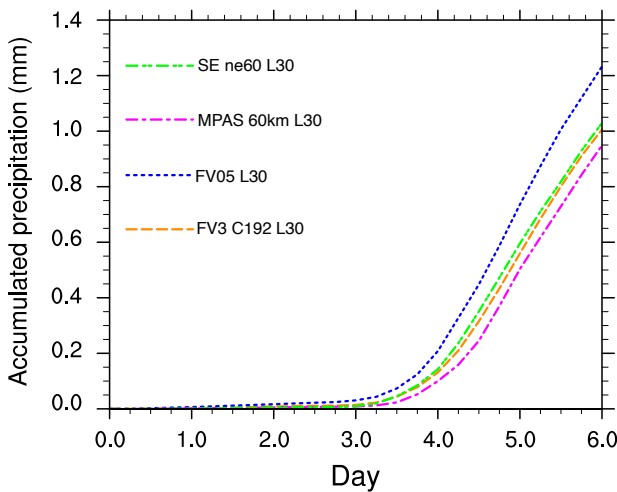

**Figure 11.** Time series of the accumulated precipitation integrated between $60 - 300°$ E and $0 - 90°$ N.

Figure 10 illustrates each model's MSLP at day 5 as the wave is forced over M2. All models exhibit qualitative agreement in the overall structure of the low and high-pressure systems. The most obvious difference is that the MPAS high-pressure systems with MSLP values over 1010 hPa occupy visibly larger areas.

## 5.2 Precipitation and diabatic forcing

It was argued by Chen and Knutson (2008) that parameterizations of large-scale precipitation are best understood as an area-average of the precipitation within a grid cell. Under this interpretation, different dynamical cores with comparable nominal grid spacings should have similar precipitation statistics, e.g., when assessing the accumulated precipitation. This holds even when point-wise convergence is not observed within a particular dynamical core as the grid spacing decreases.

Therefore, we treat the precipitation rate from the Kessler physics routine as an area average over a grid cell. Using this interpretation, the area integrals of the precipitation rate over a selected region should be comparable even when there are significant differences between the precipitation rates at individual grid points across the dynamical cores. Figure 11 shows a time series of the accumulated precipitation integrated between $60°$E and $300°$E in the northern hemisphere. The accumulation in the FV dynamical core is notably higher than the accumulation in the other dynamical cores. This holds even before day 3, when the precipitation bands along the developing frontal zones start to form. In FV, stationary orographic rain over the mountain tops is already present before hour 12. Other dynamical cores, like FV3, start the stationary orographic rain around hour 36. The reasons for these differences are not entirely clear. They are likely linked to the FV diffusion characteristics, which also caused the time evolution of the integrated EKE and minimum MSLP to differ in Fig. 8.

Figure 12 compares the longitude-height cross sections at $45°$ N of the temperature anomaly, the vertical pressure velocity, and the cloud liquid water mixing ratio as the Wave 1 precipitation band travels across the downwind slope of M2 at day 5.

In particular, Figs. 12a-d show the temperature perturbation in the four dynamical cores. Despite the grid-scale differences between models in Fig. 9, the temperature structure over the mountain is qualitatively similar across dynamical cores. The MPAS model exhibits the largest deviation from the base temperature profile. Figures 12e-h illustrate the vertical pressure velocities over the mountain at day 5. The color range for $\omega$ deliberately saturates to highlight the spatial patterns and match the color scheme of Fig. 7 (at day 4) while not displaying the actual $\omega$ minima and maxima at day 5. Overall, the vertical patterns of $\omega$ are qualitatively similar in all dynamical cores but with differences in the peak magnitudes. The figure suggests that MPAS exhibits the most intense up- and downdrafts, closely followed by FV3, which mimics the relative strength of the temperature anomalies in Figs. 12c,d. The vertical velocity and temperature anomaly patterns and magnitudes are tightly connected to the precipitation rates of the baroclinic rainbands at $140°$ and $210°$ (= $150°$ W in Fig. 9). Intense updraft areas are co-located with the rain bands. Therefore, the varying magnitudes of the updrafts help explain the local differences in the precipitation rates.

Figure 12i-l displays the distributions of the cloud liquid water mixing ratios that serve as the reservoir for rain water and precipitation in the Kessler warm-rain physics scheme. The cloud liquid water patterns broadly resemble each other, but small-scale details vary. For example, the maximum cloud liquid water mixing ratios in MPAS near $140°$ and $210°$ are located at lower altitudes under 6 km. In contrast, the peak cloud water regions in SE, FV, and FV3 are mostly found at heights of around 9-10 km. However, this might not explain the precipitation differences as the majority of the precipitation forms below 6 km. The latter is indirectly depicted by the positive temperature anomaly patterns which also capture the diabatic heating effects of precipitation. The positive temperature anomaly maxima near the rain bands at $140°$ and $210°$ are confined to regions under 6 km. MPAS exhibits the largest heating signals among the four dynamical cores. It is also interesting to observe that FV3 develops two low-lying and small cloud water clusters near $125°$ and $195°$. These are sensitive to the numerical diffusion settings in FV3 and do not appear in more diffusive FV3 configurations (not shown). As an aside, the FV simulations have difficulty keeping the cloud liquid water mixing ratio pattern compact near the $210°$ rain band.

## 5.3 Additional application aspects: Physics-dynamics coupling

The following brief discussion focuses on the physics-dynamics coupling strategy in SE and highlights an additional application area of the test case. The discussion refers back to the various physics-dynamics coupling choices for the SE model available in CESM 2.1.3 and CESM 2.2. Here, we shed light on the CESM 2.1.3 default "hybrid" coupling strategy, which was not used for the other plots in this paper.

The hybrid physics-dynamics coupling strategy in SE uses sudden adjustments of the moisture and mass fields after the physics time step (900 s in our case) and the dribbling strategy for all other physical forcings. In the chosen example at the nominal $0.5°$ resolution, SE's subcycled dynamics time step is 150 s. However, this strategy triggers spurious numerical noise (ringing) in SE, which we analyze via the vertical pressure velocity. Figure 13 shows snapshots of the 850 hPa vertical pressure velocity $\omega$ at day 5 for all four dynamical cores. All dynamical cores show small-scale gravity wave activity caused by the mountains. These are physical waves and not the focus here. Note that we saturate the color scale to draw attention to numerical artifacts. These are otherwise difficult to detect.

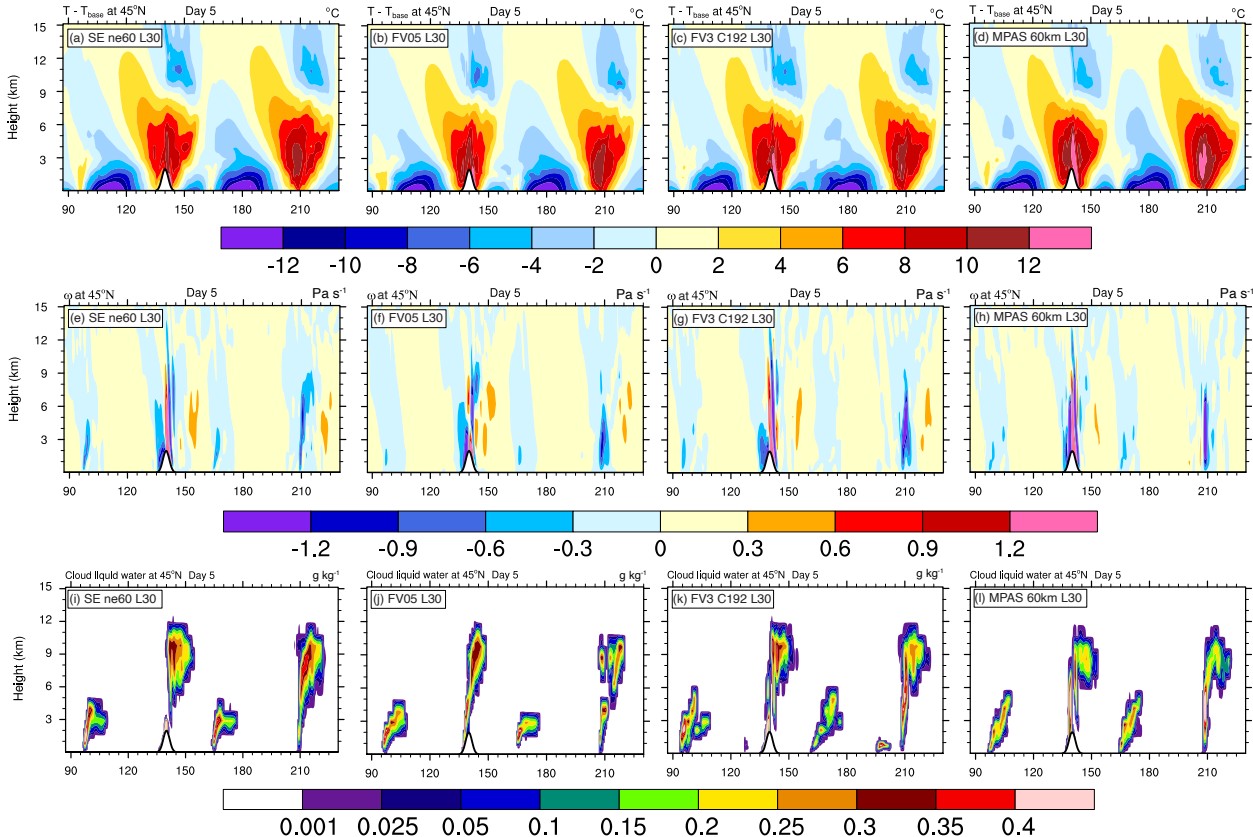

**Figure 12.** Intercomparison of longitude-height cross-sections of the (top) temperature perturbation and (middle) vertical pressure velocity $\omega$ and (bottom) cloud liquid water mixing ratio at day 5. The columns correspond to each dynamical core with a nominal $0.5°$ grid spacing. Latitude is constant at $45°$ N in SE, FV3, and MPAS, and at $44.88°$ N in FV (the closest grid point to $45°$ N). The outline of mountain M2 is shown near $140°$ E along the x-axis (longitudes).

Figure 13a demonstrates the presence of grid-scale oscillations in SE, which become more severe as the precipitation bands mature and the diabatic forcings get stronger. The oscillations appear in concentric circles and likely originate from small hotspots with strong diabatic forcing, such as grid-point storms. The magnitude of the numerical noise is small compared to the vertical velocities caused by the baroclinic wave and the mountain-generated gravity waves. However, vertical velocities are tightly coupled to cloud and rainfall characteristics. Any numerical interference in this relationship is undesirable and could lead to artificial responses in the physical parameterization.

The grid-scale oscillations occur due to the sudden moisture adjustments present with SE's hybrid coupling option. These oscillations are characteristic of SE's numerical approach, which utilizes a continuous Galerkin technique for the horizontal discretization. This phenomenon in models with local or global spectral numerical schemes is also known as Runge's phenomenon or Gibbs ringing. It resembles a shock wave that appears when large and unbalanced physical forcings are added to the rather

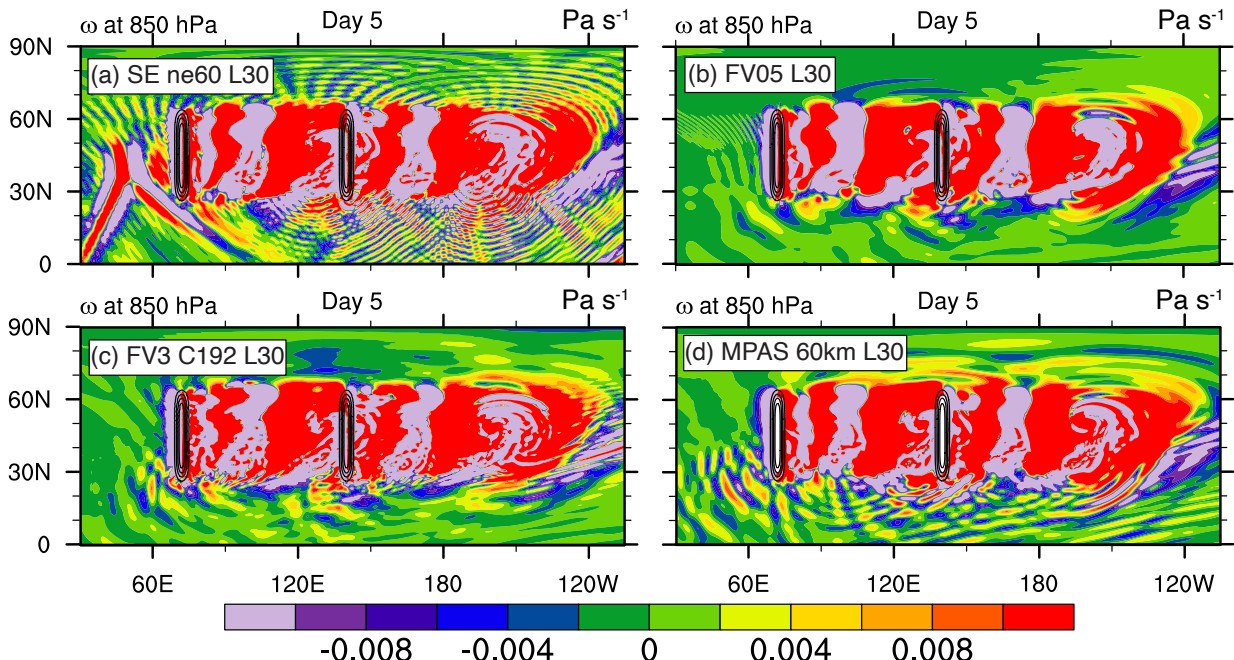

**Figure 13.** Latitude-longitude cross sections of the 850 hPa vertical pressure velocity $\omega$ from moist model simulations with a nominal $0.5°$ grid spacing in the (a) SE (with se_ftype=2), (b) FV, (c) FV3, and (d) MPAS. The color range saturates to highlight the numerical ringing/noise.

balanced motions in the dynamical core. The SE dynamical core is a highly accurate model with very low intrinsic dissipation. It becomes apparent that the explicitly added diffusion mechanisms in the SE dynamical core are insufficient to suppress these oscillations. Therefore, this noise is also tightly linked to the implicit-numerical and explicitly-added diffusion characteristics of dynamical cores. In contrast, FV (Neale et al., 2010) and FV3 (Harris et al., 2021) perform lump adjustments of prognostic fields, but their implicit-numerical and explicitly-added diffusion characteristics do not exhibit grid-scale oscillation. Similarly, the more complex strategy used by MPAS (Klemp et al., 2007), which addresses the challenges of non-hydrostatic simulations, prevents these oscillations from developing.

When dribbling is used as SE's physics-dynamics coupling strategy so that all physics tendencies are dribbled in with the subcyled 150 s dynamics time step, the spurious oscillations disappear. This was, for example, demonstrated in Hughes and Jablonowski (2021), who compared the default hybrid coupling strategy to an SE configuration with identical physics and dynamics time steps of 150 s. Reducing the time step length in the physics reduces the strength of the physical forcings. This leads to more gentle adjustments of the dynamical core and avoids spurious oscillations.

Our analysis indicates that there is a downside to using the SE default hybrid physics-dynamics coupling strategy, which was also reported for more complex, but still idealized, SE simulations in Thatcher and Jablonowski (2016). The damping of oscillations that occurs when dribbling is used is qualitatively similar to comparisons found in Gross et al. (2018). Therefore,

a direct comparison between SE simulations with hybrid and dribbled coupling is omitted. However, the dribbling strategy is also not free of numerical artifacts. Dividing the tracer tendencies for the mass quantities across dynamics substeps can result in negative tracer values, such as negative moisture (Lauritzen, personal communication, 2022). Although this is a rare circumstance, negative moisture values are unphysical and lead to problems in the physical parameterization schemes if

not filtered out beforehand via tracer mass fixers. More studies will be needed to diagnose the ringing phenomenon in more complex model configurations and determine the best way to mitigate it.

As an aside, Fig. 13a also demonstrates another unique behavior of the SE dynamical core. When comparing the vertical pressure velocity fields, SE shows very different patterns between $30° − 90°$ E just west of mountain M1. This is the signature of a horizontally traveling hydrostatic acoustic mode (a "Lamb wave"), which gets initially excited in all dynamical cores due

to the slight imbalance of the initial fields near the topography. The FV, FV3, and MPAS dynamical cores damp out the Lamb wave efficiently after about 1-2 days. However, this is not the case in SE, where the Lamb wave propagates persistently around the sphere with a phase speed of about 330 m s$^{-1}$. This topic will be discussed in a future paper.

## 6   Conclusions

This paper enhances the suite of idealized test cases for the dynamical cores of AGCMS in spherical geometry. It is the first

dynamical core test case that combines a complex initial flow field, such as the base condition for a baroclinically unstable wave with varying stratification and vertical wind shear, with idealized topographic barriers on a rotating regular-size earth. Both dry and idealized moist test configurations are suggested for pressure-based and height-based dynamical cores. The moist configuration utilizes a warm-rain Kessler physics parameterization that triggers precipitation and provides diabatic forcing. The test accommodates the portfolio of hydrostatic and non-hydrostatic as well as shallow-atmosphere and deep-atmosphere

dynamical core designs. In particular, we add an analytically-defined topography profile to an existing baroclinic wave base state. This necessitates re-balancing the initial conditions with a particular focus on the surface pressure and vertical velocity fields. The latter is only needed for non-hydrostatic models. The resulting initial conditions are well-balanced but not a steady-state solution. The topography field acts as the trigger for rapidly growing baroclinic waves over several days.

The test case provides a controlled environment that serves two purposes. First, it can be used as a model assessment, debug-

ging, or tuning tool by model developers who need to assess the inclusion of topography and the chosen vertical coordinate in a dynamical core. This informs numerical design decisions for dynamical cores and their physics-dynamics coupling strategy and contributes to dynamical core model intercomparisons. Second, the test case also serves the atmospheric dynamics science community. It is a tool in the atmospheric dynamics toolbox and sheds, for example, light on the impact of mountains on the general circulation. All mountain shapes can be accommodated as long as they are prescribed via analytic functions. An analyt-

ical solution does not exist. However, high-resolution reference solutions and dynamical core intercomparisons can be used to gain confidence in the model simulations. This is straightforward in dry configurations that converge with increasing resolution before nonlinear wave-breaking and mixing processes set in after day 6.5. However, moist configurations are impacted by the nonlinear forcing from the stationary orographic rain and, most importantly, the precipitation along the frontal zones from day

3 onwards. This leads to an increased spread in the model simulations that typically exhibit wave breaking after day 5 for the chosen topographic profile.

We illustrated the characteristics and capabilities of the test case via example simulations with various dynamical cores, which are available at NCAR. These are the SE and FV dynamical cores of the CESM 2.1.3 model framework, the FV3 of the CESM 2.2 model framework, and the standalone distribution of MPAS version 7. The dynamical characteristics of the topographically triggered baroclinic waves were studied and a model intercomparisons was performed. Real-world flow phenomena and mountain shapes were used to inspire the selection of the flow parameters, such as the shape of the two chosen ridge mountains. These triggered baroclinic waves that have similarities with atmospheric rivers. The chosen examples showcase the potential use cases of this test case. Besides serving as a debugging tool, we briefly discussed the impact of diffusion on the flow and precipitation characteristics. Furthermore, the test revealed physics-dynamics coupling problems in the SE dynamical core. It also led to the discovery of an acoustic mode in the SE model that persistently propagates in the horizontal direction without much damping. Overall, the overall flow patterns in the dynamical core simulations resembled each other when evaluating selected quantitative metrics. However, the details can differ greatly at the local level. The results suggest that FV's diffusion characteristic noticeably impacted how the wave evolved.

There are several future directions for this research. First, the underlying atmospheric base state has both a shallow-atmosphere and a deep-atmosphere variant. So far, we have tested the shallow-atmosphere variant, but we are intrigued to apply the test to deep-atmosphere dynamical cores, which are becoming popular. Second, the test can also utilize a reduced-radius configuration to trigger non-hydrostatic model responses more easily. However, Skamarock et al. (2021) identified that care needs to be taken so that statically unstable regions do not develop when the radius reduction factor $X$ is large. The authors of that study suggested slight adjustments of the base flow to prevent the formation of unstable regions. Third, the model integrations with the SE dynamical core revealed that the dynamical core preserves a rapidly propagating acoustic-gravity (Lamb) mode. Although this mode is initially present in other dynamical cores due to slight imbalances of the initial conditions, it is rapidly damped in all tested dynamical cores except SE. A systematic analysis of this phenomenon is deferred to a future publication. In addition, it will be interesting to systematically investigate the impact of implicit numerical and explicitly added diffusion on the evolution of the geographically triggered baroclinic waves.

## Appendix A: Description of the Kessler Physics Parameterization

This appendix reviews the "Kessler physics" processes (Kessler, 1969), which represent a warm-rain cloud microphysics scheme without an ice phase. The recommended method for adding the Kessler physics processes to a dynamical core is to use and adapt the provided `kessler.F90` Fortran file. This Fortran template routine is available in the Zenodo archive that accompanies this publication (see the Code Availability section for the web link). The routine was originally developed for the DCMIP modeling groups in 2016. It was based on the Kessler physics routine listed in Appendix C in Klemp et al. (2015). The Kessler parameterization is often available as a switch-on option in many existing code bases, such as CESM, MPAS, the

Weather Research and Forecasting (WRF) model (Skamarock et al., 2008), and in GFDL's "Solo" configuration of the FV3 dynamical core (available via GitHub).

Some variants of the Kessler physics scheme exist in the literature. Here, we document our chosen variant that closely resembles the implementation in Klemp and Wilhelmson (1978) and Klemp et al. (2015), and was furthermore utilized for the
DCMIP dynamical core intercomparison in 2016 (Ullrich et al., 2016; Zarzycki et al., 2017). A similar implementation of the Kessler processes is also detailed in Durran and Klemp (1983) (see Appendix 2). The scheme utilizes three prognostic moisture variables: the dry mixing ratios for water vapor $m_v$, cloud water $m_c$, and rain water $m_r$. The included microphysical processes are (a) the production, sedimentation, and evaporation of rain water, (b) the collection (accretion) and autoconversion of cloud water, and (c) and the production of cloud water from condensation. The time tendencies for the potential temperature $\theta$ and
the three water species are then expressed via the following equation set:

$$\frac{d\theta}{dt} = \frac{L}{\tilde{c}_p\Pi}\Big(C_{cond} - E_r\Big) \tag{A1}$$

$$\frac{dm_v}{dt} = -C_{cond} + E_r \tag{A2}$$

$$\frac{dm_c}{dt} = C_{cond} - A_r - C_r \tag{A3}$$

$$\frac{dm_r}{dt} = -E_r + A_r + C_r - S, \tag{A4}$$

where $\Pi$ is the Exner function, $L = 2.5 \times 10^6$ J K$^{-1}$ is the latent heat of vaporization, $\tilde{c}_p = 1003$ J K$^{-1}$ is the specific heat at constant pressure as utilized in Klemp et al. (2015), and p is the moist pressure (see also Eq. (B5)). The symbol $C_{cond}$ denotes the condensation rate (defined to be positive in case of condensation), $E_r$ represents the rain water evaporation rate, $A_r$ symbolizes the autoconversion rate of cloud water to rain water, $C_r$ stands for the collection rate of rain water, and $S$ displays the sedimentation rate. Contrary to the notation in Klemp and Wilhelmson (1978), Klemp et al. (2015), and Ullrich
et al. (2016), we denote the dry mixing ratios for the water species with the symbol $m$ instead of the symbol $q$. The symbol $q$ is typically used for moist mixing ratios in the literature. The general conversion equations between dry and moist mixing ratios are given by

$$q_X = \frac{m_X}{1 + (m_v + m_c + m_r)} \tag{A5}$$

$$m_X = \frac{q_X}{1 - (q_v + q_c + q_r)} \tag{A6}$$

where the subscript $X$ is a placeholder for $v, c, r$. In case a dynamical core uses moist mixing ratios, it is paramount to convert the moist mixing ratios to dry mixing ratios before the Kessler physics routine is called. After the Kessler physics routine updates the dry mixing ratios, they must be converted back to their moist equivalents for the subsequent dynamical core computations. This moist/dry conversion requires knowledge about the design of the dynamical core. Some dynamical cores only use the water vapor contribution to compute the moist mixing ratios and leave out the contributions from the
condensates. If this is the case the conversions (A5) and (A6) simplify and need to utilize $m_c = m_r = q_c = q_r = 0$. Another notational difference is the use of the symbol $C_{cond}$ for the condensation rate, which is equivalent to the term $-dq_{vs}/dt$ (equal

to $-dm_{vs}/dt$) in Klemp and Wilhelmson (1978) or Klemp et al. (2015). The computation of the saturation mixing ratio $m_{vs}$ uses Tetens' formula, which is shown in Eq. (B8).

The autoconvection rate $A_r$ and collection rate $C_r$ follow the Kessler parameterization and are defined by

$$A_r = k_1 (m_c - a_r) \tag{A7}$$

$$C_r = k_2 m_c m_r^{0.875} \tag{A8}$$

with $k_1 = 0.001 \text{ s}^{-1}$, $a_r = 0.001 \text{ g g}^{-1}$ and $k_2 = 2.2 \text{ s}^{-1}$. In addition, $\Delta m_r$ is defined as the rain production term

$$\Delta m_r = m_c^n - \frac{m_c^n - \Delta t \max(A_r, 0)}{1 + \Delta t C_r} \tag{A9}$$

with the physics time step $\Delta t$ which is typically subcycled (see the additional explanations below). The sedimentation rate $S$ is described by

$$S = \frac{1}{\rho_d} \frac{d(\rho_d V_r m_r)}{dz} \tag{A10}$$

as shown by Eq. (2.9b) in Klemp and Wilhelmson (1978) which utilizes the rain water terminal velocity $V_r$ in units of m s$^{-1}$

$$V_r = 36.34 \, (\rho_{gm} m_r)^{0.1346} \sqrt{\frac{\rho_0}{\rho_d}}. \tag{A11}$$

The expression for $V_r$ corresponds to Eq. (2.15) in Klemp and Wilhelmson (1978) where $\rho_{gm}$ symbolizes the density of dry air in units of g cm$^{-3}$, and $\rho_0$ and $\rho_d$ denote the density of dry air at the lowest model level and the chosen vertical position, respectively. This term is discretized via an upstream finite-difference method. The implementation details are shown in Appendix C in Klemp et al. (2015) and the `kessler.F90` Fortran file.

These processes are then used to provide a temporarily updated rain water mixing ratio given by

$$m_r^* = \max(m_r^n + \Delta m_r + S \Delta t, 0) \tag{A12}$$

where $n$ is the current time index. The final update of $m_r$ takes the rain water evaporation into account. The rain water evaporation rate (Eq. (2.14a) in Klemp and Wilhelmson (1978)) in units of s$^{-1}$ is

$$E_r = \frac{1}{\rho_{gm}} \frac{\left(1 - \frac{m_v}{m_{vs}}\right) C (\rho_{gm} m_r^*)^{0.525}}{5.4 \times 10^5 + \frac{2.55 \times 10^6}{p_{hPa} m_{vs}}} \tag{A13}$$

which utilizes the ventilation coefficient C

$$C = 1.6 + 124.9 (\rho_{gm} m_r^*)^{0.2046}, \tag{A14}$$

and the pressure $p_{hPa}$ in units of hPa. Condensation is triggered if the water vapor mixing ratio $m_v$ exceeds the saturation mixing ratio $m_{vs}$. In this case, the condensation rate is positive and utilizes the equation

$$C_{cond} = \frac{1}{\Delta t} \frac{m_v - m_{vs}}{1 + m_{vs} \frac{17.27 \times 237.3 L}{\tilde{c}_p (T - 36)^2}} \tag{A15}$$

which is also shown as an update equation (without $\Delta t$) in Durran and Klemp (1983) (their Eq. (A14)). The factor $17.27 \times 237.3$ is about 4098, which is close to the factor $17.27 \times 237 \approx 4093$ listed in Klemp and Wilhelmson (1978) and Klemp et al. (2015).

**Implementation details**

It is essential to recognize that the provided `kessler.F90` Fortran file expects vertical column data that start at the lowest model level near the surface and extend upward. If a dynamical core counts the levels from the top down, the levels need to be reordered before the Kessler parameterization is called. In addition, the sedimentation process is discretized via an upstream finite-difference approach, as mentioned above, which needs to obey numerical stability constraints. Therefore, the Kessler

physics processes must be subcycled in time unless the physics time step is short enough to guarantee numerical stability. The subcycling is implemented in the provided Fortran routine. It is not part of the Kessler implementation shown in Klemp et al. (2015) that implicitly assumes that the physics time step is numerically stable. The Kessler physics routine computes five output variables. These are the updated water vapor, liquid water, and rain water mixing ratios at the future time step (after the duration of a full physics time step), the updated potential temperature, and the averaged precipitation rate. The precipitation

rate

$$Precipitation = \frac{\rho_0 \, m_{r0} \, V_{r0}}{\rho_{water}} \tag{A16}$$

represents the sedimentation from the lowest model level indicated by the subscript 0. It accounts for the accumulated precipitation (in meters of water per second) over a subcycled time step with the density of water $\rho_{water} = 1000$ kg m$^{-3}$. The precipitation rate differs from substep to substep and must be averaged. All precipitation rates are therefore accumulated over

the full physics time step and then divided by number of substeps to compute the average precipitation rate as an output quantity. There are also other implementation details that affect the accuracy of the parameterization. For example, the computation of the rain water evaporation rate $E_r$ and the updated water substances must ensure that the mixing ratios do not become negative. This necessitates the use of max and min functions as well as limiters. Therefore, we recommend using the provided Fortran routine or closely reviewing the implementation details to avoid any numerical difficulties.

**Appendix B: Description of the Initial State**

This appendix presents selected equations for the moist initial state in a shallow-atmosphere configuration introduced in Ullrich et al. (2016). The equations containing the adjustments for the topographic profile are discussed in Sect. 2.2. These topographic adjustments enter the equations via the height variable. Users of deep-atmosphere dynamical cores should review the needed slight adjustments outlined in Ullrich et al. (2014). The equations below do not formally specify a dependence on the longitude.

However, this dependence is implicit as the height $z$ and pressure $p$ along a model level are now functions of both horizontal directions over the topography. Table B1 lists all parameters and physical constants for the initial conditions, including an optional small-earth scaling factor $X$. It is set to an unscaled value of $X = 1$ here but could be varied in future work to trigger non-hydrostatic model responses. Note that such scaling reduces the earth's radius and speeds up the earth's rotation

**Table B1.** Parameters and physical constants for the initial conditions.

| Variable Name | Variable Description | Value |
|:---:|:---|:---|
| $X$ | 1 | Reduced-size planet scaling factor |
| $a$ | $6.37122 \times 10^6 \text{ m} \cdot X^{-1}$ | Scaled radius of the earth |
| $\Omega$ | $2\pi \left(86164 \text{ s}\right)^{-1} \cdot X$ | Scaled angular speed of the earth |
| $g$ | $9.80616 \text{ m s}^{-2}$ | Gravity |
| $R_d$ | $287 \text{ J kg}^{-1} \text{ K}^{-1}$ | Gas constant for dry air |
| $p_0$ | $10^5 \text{ Pa}$ | Reference pressure |
| $b$ | 2 | Jet half-width parameter |
| $K$ | 3 | Power used for temperature field |
| $T_{\text{E}}$ | 310 K | Reference surface temperature at the equator |
| $T_{\text{P}}$ | 240 K | Reference surface temperature at the poles |
| $\Gamma$ | $0.005 \text{ K m}^{-1}$ | Temperature lapse rate |
| $\phi_{\text{w}}$ | $4\pi/18$ | Specific humidity latitudinal width parameter in radians |
| $p_{\text{w}}$ | $3.4 \times 10^4 \text{ Pa}$ | Specific humidity vertical pressure width parameter |
| $q_0$ | $0.018 \text{ kg kg}^{-1}$ | Maximum specific humidity |
| $q_t$ | $0 \text{ kg kg}^{-1}$ | Specific humidity above artificial tropopause |
| $p_t$ | $1.5 \times 10^4 \text{ Pa}$ | Pressure at artificial tropopause |
| $L$ | $2.5 \times 10^6 \text{ J kg}^{-1}$ | Latent heat of vaporization |
| $c_p$ | $1004.5 \text{ J kg}^{-1} \text{ K}^{-1}$ | Specific heat at constant pressure |
| $R_v$ | $461.5 \text{ J kg}^{-1} \text{ K}^{-1}$ | Gas constant for water vapor |
| $\Pi$ | $\left(p/p_0\right)^{R_d/c_P}$ | Exner function, $p$ is the pressure of the moist air in Pa |

simultaneously to keep the Rossby number constant. As further explained in Ullrich et al. (2016), other changes are also needed
for reduced-radius experiments. The implementation details for the initial conditions and the CESM 2.2 (FV3), CESM 2.1.3
(FV and SE), and MPAS simulations are provided in Appendix C. If a chosen dynamical core uses slightly different values for
the physical constants, we recommend using the model's defaults to provide an internally consistent initialization.

## B1    Temperature base state

The temperature equation is a particular form of the temperature family given in Staniforth and White (2011), which the
interested reader can consult for explanations of how these functional forms were chosen. This has a variation in the meridional
direction, which is determined by the parameter $K$:

$$I_T(\phi) = \left(\cos\phi\right)^K - \frac{K}{K+2}\left(\cos\phi\right)^{K+2}.$$

In addition, two height-dependent functions are needed. They are given by

$$\tau_1(z) = \frac{1}{T_0} \exp\left(\frac{\Gamma z}{T_0}\right) + \left(\frac{T_0 - T_P}{T_0 T_P}\right) \left[1 - 2\left(\frac{zg}{bR_d T_0}\right)^2\right] \exp\left[-\left(\frac{zg}{bR_d T_0}\right)^2\right]$$

$$\tau_2(z) = \frac{K+2}{2} \left(\frac{T_E - T_P}{T_E T_P}\right) \left[1 - 2\left(\frac{zg}{bR_d T_0}\right)^2\right] \exp\left[-\left(\frac{zg}{bR_d T_0}\right)^2\right]$$

with the vertical lapse rate $\Gamma$, and the meridional temperature gradient. The latter is expressed via the equatorial and polar temperature parameters $T_E$ and $T_P$, respectively. The parameter $T_0 = \frac{1}{2}(T_E + T_P)$ denotes the arithmetic mean.

To incorporate water vapor into our base state, we first specify the virtual temperature $T_v$ as follows

$$T_v(\phi, z) = \frac{1}{\tau_1(z) - \tau_2(z) I_T(\phi)}$$

which obeys the thermal wind balance. The prognostic temperature initialization $T$ is therefore

$$T = \frac{T_v}{1 + M_v q_v} \tag{B1}$$

with $M_v = 0.608$. The symbol $q_v$ denotes the specific humidity, as explained below.

## B2   Zonal wind base state

The zonal wind and virtual temperature are connected via the thermal wind balance. The dependence on $T_v$ is sequestered in the auxiliary quantity:

$$U(\phi, z) = \frac{gK}{a} \tau_{\text{int},2}(z) \left[(\cos\phi)^{K-1} - (\cos\phi)^{K+1}\right] T_v(\phi, z)$$

from which we can derive the prognostic zonal wind initialization as

$$u(\phi, z) = -\Omega a \cos(\phi) + \sqrt{(\Omega a \cos(\phi))^2 + a \cos(\phi) U(\phi, z)}. \tag{B2}$$

## B3   Meridional wind base state

The meridional wind is set to zero, with

$$v \equiv 0 \text{ m s}^{-1}. \tag{B3}$$

## B4   Pressure and density base state

The pressure distribution is determined by

$$p(\phi, z) = p_0 \exp\left[-\frac{g}{R_d}\left(\tau_{\text{int},1}(z) - \tau_{\text{int},2}(z) I_T(\phi)\right)\right] \tag{B4}$$

where the integrals of the height-dependent functional forms for temperature are given by

$$\tau_{\text{int},1}(z) = \frac{1}{\Gamma}\left[\exp\left(\frac{\Gamma z}{T_0}\right) - 1\right] + z\left(\frac{T_0 - T_{\text{P}}}{T_0 T_{\text{P}}}\right)\exp\left[-\left(\frac{zg}{bR_d T_0}\right)^2\right]$$

and

$$\tau_{\text{int},2} = \frac{K+2}{2}\left(\frac{T_{\text{E}} - T_{\text{P}}}{T_{\text{E}} T_{\text{P}}}\right) z \exp\left[-\left(\frac{zg}{bR_d T_0}\right)^2\right].$$

The symbol $p$ denotes the pressure of the moist air. The surface pressure $p_s$ is then provided when plugging in the topographic height $z_s$ (Eq. (1)) into Eq. (B4) as shown in Eq. (2). Using the virtual temperature equation, the density of the moist air is determined by the ideal gas law

$$\rho = \frac{p}{R_d T_v}. \tag{B5}$$

### B5    Base states for the moisture variables

With the help of the auxiliary quantity $\eta$

$$\eta(\phi, z) = \frac{p(\phi, z)}{p_0},$$

the initial distribution of the specific humidity is given by

$$q_v(\phi, \eta) = \begin{cases} q_0 \exp\left[-\left(\frac{\phi}{\phi_w}\right)^4\right]\exp\left[-\left(\frac{(\eta-1)p_0}{p_w}\right)^2\right], & \text{if } \eta > p_t/p_0 \\ q_t, & \text{otherwise.} \end{cases} \tag{B6}$$

The specific humidity corresponds to a wet mixing ratio for water vapor. The initial values for the wet mixing ratios of cloud
water $q_c$ and rain water $q_r$ are set to zero. This means the initial values for the dry mixing ratios of cloud water $m_c$ and rain water $m_r$ are also zero. If a dynamical core utilizes the dry mixing ratio for water vapor $m_v$ instead of the specific humidity $q_v$, the conversion shown in Eq. (A6) needs to be applied. The conversion can either involve just the vapor contribution or also the condensates which depends on the design of the dynamical core.

### B6    Relative humidity

The relative humidity distribution makes use of Tetens' formula for the saturation mixing ratio, as also shown in Klemp and Wilhelmson (1978), Klemp et al. (2015), and Durran and Klemp (1983). Note that the Klemp et al. (2015) formulation (their Eq. (12)) contains a typographical error when stating that their pressure $\bar{p}_{eq}$ has units of hPa. The correct unit for the pressure $p$ in the denominator is Pa, as shown below. Here, we use Tetens' formula for the saturation specific humidity $q_{vs}$, which is approximately equal to the saturation mixing ratio $m_{vs}$. The formula is

$$
\quad q_{vs} \quad = \quad \frac{\epsilon}{p}\, e_0^* \exp\left(17.27\,\frac{T - 273\,\text{K}}{T - 36\,\text{K}}\right) \tag{B7}
$$

$$
= \quad \frac{380\,\text{Pa}}{p}\exp\left(17.27\,\frac{T - 273\,\text{K}}{T - 36\,\text{K}}\right) \tag{B8}
$$

$$
\approx \quad m_{vs}
$$

where the units of $p$ and $T$ are Pa and K, respectively. For illustration purposes, Eq. (B7) also lists the saturation vapor pressure $e_0^* = 610.78$ Pa at the temperature triple point $T_{00} = 273.16$ K and the symbol $\epsilon = R_d R_v^{-1} \approx 0.622$ which denotes the ratio of the gas constant for dry air $R_d$ to that for water vapor with $R_v$. This explains the physical meaning of the constant 380 Pa in Eq. (B8). The relative humidity $RH$ can then be defined as

$$\mathrm{RH} = 100\% \cdot \frac{q_v}{q_{vs}}$$

### Appendix C: Implementation Details for the CESM2 Dynamical Cores and MPAS

We recommend using the default physical constants as implemented in a chosen dynamical core and ideally in-lining the initialization routine in the codebase of the chosen model. This was the initialization strategy for the CESM 2 and MPAS dynamical cores, and all code modifications are provided in Hughes and Jablonowski (2022). In CESM 2, we made use of CESM's "Simpler Models" framework, which invokes the Kessler physics routine described in Appendix A and the analytic initialization of the moist baroclinic wave (the Ullrich et al. (2016) default without topography). We utilize the CESM compset "FKESSLER" and a CAM namelist entry for the variable `analytic_ic_type = 'moist_baroclinic_wave_dcmip2016'`. The CESM 2 code change then augments the existing initialization for the baroclinic wave and adds the topographic changes via a swap of the CAM routine `ic_baroclinic.F90`. Note that the routine `ic_baroclinic.F90` also accommodates the dry variant of the baroclinic wave, which can be selected via the adiabatic compset "FADIAB", the configure command `./xmlchange --append --file env_build.xml --id CAM_CONFIG_OPTS --val="--analytic_ic"` to activate the analytic in-lined initialization and the alternative namelist option `analytic_ic_type = 'dry_baroclinic_wave_dcmip2016'`. These settings initialize the dry configuration with $q = 0$ and do not activate any physical parameterizations. All key namelist entries are provided in Tables C1-C4. The CESM values for the physical constants are listed in Table B1. For the MPAS simulations, the default physical constants of the MPAS stand-alone distribution were used (Jacobsen et al., 2019). MPAS provides the implementation of the Kessler warm-rain microphysics routine, which can be activated via a namelist option as shown in Table C4. In addition, the initialization routine for the moist baroclinic wave with topography was added to MPAS' existing framework for idealized test cases via a code change.

All dynamical core simulations are run with 30 model levels and use model tops near 2 hPa (SE, FV, FV3) and 8 hPa (MPAS). These model tops lie between 30-35 km for the provided temperature structure. The positions of the hybrid pressure-based model level used for SE, FV, and FV3 are listed in Reed and Jablonowski (2012) and are recommended to users of this test case. These are the default levels in CESM 2.1.3 and CESM 2.2 once the compset FKESSLER is invoked. For MPAS, we use the 30 default levels for MPAS' idealized testing framework. Most simulations presented in this study are run with a nominal $0.5°$ (about 50 km) grid spacing, which corresponds to the grid resolution settings ne60 (SE), FV05 (FV), C192 (FV3), and 60 km (MPAS). These identifiers are used as labels in the figures and correspond to the time step and diffusion settings quoted below.

Table C1 contains the key namelist parameters to replicate our SE model integrations. The time steps used by the SE dynamical core are $\Delta_{\mathrm{phys}} = 900$ s, $\Delta_{\mathrm{vertical\ remap}} = \Delta_{\mathrm{phys}}/2 = 450$ s, $\Delta_{\mathrm{dynamics}} = \Delta_{\mathrm{vertical\ remap}}/3 = 150$ s, and $\Delta_{\mathrm{hyperviscosity}} =$

**Table C1.** Key CESM 2.1.3 namelist parameters used in nominal $0.5^\circ$ SE model integrations.

| Namelist Parameter | Value |
| --- | --- |
| analytic_ic_type | 'moist_baroclinic_wave_dcmip2016' |
| se_ftype | 0 |
| se_hypervis_on_plevs | .true. |
| se_hypervis_subcycle | 3 |
| se_hypervis_subcycle_q | 1 |
| se_limiter_option | 8 |
| se_ne | 60 |
| se_nsplit | 2 |
| se_qsplit | 1 |
| se_rsplit | 3 |
| se_tstep_type | 4 |
| se_vert_remap_q_alg | 1 |
| se_nu | 0.40E+14 |
| se_nu_div | 0.10E+15 |
| se_nu_p | 0.10E+15 |
| se_nu_top | 2.5e5 |

$\Delta_{\text{dynamics}}/3 = 50$ s as, for example, explained in Lauritzen et al. (2018). The se_nu_XX parameters denote diffusion coefficients, which are resolution-dependent.

Table C2 contains the key namelist parameters to replicate the FV model integrations. The time steps used by the FV dynamical core are $\Delta_{\text{phys}} = 900$ s, $\Delta_{\text{vertical remap}} = \Delta_{\text{phys}}/2 = 450$ s, $\Delta_{\text{tracer}} = \Delta_{\text{vertical remap}} = 450$ s, and $\Delta_{\text{dynamics}} = \Delta_{\text{tracer}}/4 =$
112.5 s. The namelist entry fv_div24del2flag selects the 4th-order horizontal divergence damping mechanism. The monotonicity constraints for the horizontal advection and the vertical remap algorithm, denoted by the fv_Xord namelist entries, are called the "relaxed constraint" by Lin (2004) and denote the default settings.

Table C3 contains the key namelist parameters for the FV3 model integrations. The time steps used by the FV3 dynamical core are $\Delta_{\text{phys}} = 900$ s, $\Delta_{\text{vertical remap}} = \Delta_{\text{phys}}/2 = 450$ s, $\Delta_{\text{tracer}} = \Delta_{\text{vertical remap}} = 450$ s, and $\Delta_{\text{dynamics}} = \Delta_{\text{vertical remap}}/6 =$
75 s. The "fv3_nord = 2" setting activates the 6th-order horizontal divergence damping mechanism with the dimensionless resolution-independent coefficient fv3_d4_bg. The optional vorticity damping is not activated. The choice of the monotonicity constraint for the horizontal advection, as determined by fv3_hord_XX, picks the least diffusive option.

Table C4 contains the key namelist parameters for MPAS. The time steps used by the MPAS dynamical core are: $\Delta_{\text{phys}} = 300$s, $\Delta_{\text{dynamics}} = \Delta_{\text{phys}}/3 = 100$ s, and $\Delta_{\text{acoustic}} = \Delta_{\text{dynamics}}/2 = 50$ s. MPAS is a non-hydrostatic model, and so it ensures
numerical stability in the presence of 3D acoustic waves by handling acoustic propagation with very short time steps.

**Table C2.** Key CESM 2.1.3 namelist parameters used in nominal 0.5° FV model integrations.

| Namelist Parameter | Value |
| --- | --- |
| analytic_ic_type | 'moist_baroclinic_wave_dcmip2016' |
| fv_div24del2flag | 4 |
| fv_fft_flt | 1 |
| fv_filtcw | 0 |
| fv_nspltvrm | 2 |
| fv_nsplit | 0 |
| fv_iord | 4 |
| fv_jord | 4 |
| fv_kord | 4 |

**Table C3.** Key CESM 2.2 namelist parameters used in nominal 0.5° FV3 model integrations.

| Namelist Parameter | Value |
| --- | --- |
| analytic_ic_type | 'moist_baroclinic_wave_dcmip2016' |
| fv3_hydrostatic | .true. |
| fv3_hord_mt | 5 |
| fv3_hord_vt | 5 |
| fv3_hord_tm | 5 |
| fv3_hord_dp | −5 |
| fv3_hord_tr | 8 |
| fv3_kord_mt | 9 |
| fv3_kord_tm | −9 |
| fv3_kord_tr | 9 |
| fv3_kord_wz | 9 |
| fv3_n_split | 6 |
| fv3_k_split | 2 |
| fv3_do_vort_damp | .false. |
| fv3_nord | 2 |
| fv3_d4_bg | 0.15 |
| fv3_d2_bg | 0. |
| fv3_d2_bg_k1 | 0.15 |
| fv3_d2_bg_k2 | 0.02 |
| fv3_rf_cutoff | 750 |
| fv3_tau | 10 |

**Table C4.** Key namelist parameters used in nominal 0.5° MPAS model integrations.

| Namelist Parameter | Value |
| --- | --- |
| config_dt | 300.0 |
| config_split_dynamics_transport | true |
| config_number_of_sub_steps | 2 |
| config_dynamics_split_steps | 3 |
| config_horiz_mixing | '2d_smagorinsky' |
| config_len_disp | 60000.0 |
| config_visc4_2dsmag | 0.05 |
| config_u_vadv_order | 3 |
| config_w_vadv_order | 3 |
| config_w_adv_order | 3 |
| config_theta_vadv_order | 3 |
| config_scalar_vadv_order | 3 |
| config_theta_adv_order | 3 |
| config_scalar_adv_order | 3 |
| config_scalar_advection | true |
| config_positive_definite | false |
| config_coef_3rd_order | 0.05 |
| config_del4u_div_factor | 10.0 |
| config_apvm_upwinding | 0.5 |
| config_monotonic | true |
| config_epssm | 0.1 |
| config_smdiv | 0.1 |
| config_physics_suite | 'none' |
| config_microp_scheme | 'mp_kessler' |

*Code and data availability.* Model data, the Kessler physics routine, and source code to run the test case in the CESM and MPAS models have been uploaded to a Zenodo dataset (Hughes and Jablonowski, 2022). The collection includes the data analysis scripts that reproduce the figures from the article.

*Author contributions.* CJ suggested the study. OKH and CJ designed the specifications of the test case. OKH developed the code modifications, conducted the model experiments, and analyzed the model results. OKH and CJ jointly wrote the manuscript.

*Competing interests.* The authors declare that they have no conflict of interest.

*Acknowledgements.* This work was supported by the NOAA grant NA17OAR4320152(127) and the Department of Energy, Office of Science, grant DE-SC0023220. We would like to acknowledge high-performance computing support from Cheyenne (doi:10.5065/D6RX99HX) and the Casper data analysis server. These resources were provided by the Computational and Information Systems Laboratory of the National Center for Atmospheric Research (NCAR), sponsored by the National Science Foundation. This research was partly supported through computational resources and services provided by Advanced Research Computing at the University of Michigan, Ann Arbor. We thank Hilary Weller and the second anonymous reviewer for their helpful comments and suggestions which improved the manuscript.

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
