# Peer review of "A Mountain-Induced Moist Baroclinic Wave Test Case for the Dynamical Cores of Atmospheric General Circulation Models"

_EGUsphere, 2023_

## Author Response (AR1)

**Response to Reviewer 1, Manuscript: egusphere-2023-376**

Owen Hughes and Christiane Jablonowski

**1  Author Comments**

We thank Reviewer 1 for their greatly appreciated constructive feedback, which improved the quality of the manuscript. When referring to line number edits made by the reviewers, we refer to the revised version of the manuscript. The key changes in the revised version of the manuscript are as follows:

- We have used the sentence-level edits to improve the flow of the language in the paper.

- We have added additional references to Sect. 5.3, which place the physics-dynamics coupling difficulties observed in SE in the context of the existing literature.

- We rephrased the conclusion to delineate better that modifications to the deep-atmosphere steady state were suggested in Skamarock et al. (2021).

**2  Response to Reviewer 1**

We thank the reviewer for their kind words and detailed review.

**Comment 1:**

In 2.3 you describe the method of initialising w for non-hydrostatic models which, you say, doesn't have much impact. I would imagine that, if you did need to get rid of initial condition shocks then you would need to apply a projection to get discretely divergence free initial conditions.

**Response:** This is an interesting idea, but the implementation would be largely model-dependent. This may be especially useful for the further work we are doing on the Lamb wave, which results from the initial imbalance (although this also appears in hydrostatic models). We thank the reviewer for their potentially useful insights. However, the imbalances are small enough that we have elected not to address this point in the current publication.

**Comment 2:**

On lines 399-403 you talk about comparing area averages of precipitation. However it is not clear at this stage that this is what you are going to do. The relevance of the paragraph is not clear.

**Response:** The reviewer is correct that this paragraph is more relevant to subsection 5.2, Precipitation and diabatic forcing. The reference to Chen and Knutson (2008) provides some theoretical justification for why a summary statistic such as area-integrated precipitation is one way of assessing whether a model (in this case, FV) falls outside of the distribution of models at a particular nominal resolution when pointwise convergence cannot be expected.

**Comment 3:**

On line 480 I would say "compares" rather than "intercompares".

**Response:** This wording has been improved in the manuscript in line with this comment.

**Comment 4:**

On line 485, "the" rather than "to".

**Response:** The language has been adjusted following the comment.

**Comment 5:**

On line 502, "have" rather than "has"

> **Response:** The language has been adjusted following the comment.

**Comment 6:**

On line 503, "compact" rather than "concise".

> **Response:** The language has been adjusted following the comment.

**Comment 7:**

The paragraph including line 530 sounds important, describing how you can remove spurious oscillations. I would think that you should show evidence for this.

> **Response:** While this is an important solution, the problem with SE's coupling strategy and the resultant fix have already been demonstrated in slightly more complex test cases. Language has been added on Line 537 to direct readers to direct comparisons in these other references, which have highly similar qualitative characteristics to the comparison for our test case.

**Comment 8:**

Line 548, why is damping out a Lamb wave considered to be more desirable than maintaining a Lamb wave.

> **Response:** This paragraph does not explicitly state that it is more desirable. Historically, authors have conducted work to remove horizontal acoustic modes at model initialization (see, e.g., Washington and Baumhefner (1975)). However, the wave detected in our test case is a physical response to a numerical imbalance at initialization. Therefore, once the wave is triggered, it should be preserved. The further work we refer to in this manuscript will be concerned with why this wave is triggered and an explanation of why it is preserved in one out of four dynamical cores studied.

**Comment 9:**

Line 579 you say that you have "discussed the impact of diffusion on the flow and precipitation". You haven't discussed this much.

> **Response:** These impacts of diffusion are mentioned in the context of model intercomparisons. Lines 435-440 identify FV's diffusive mechanisms as a cause of slower wave amplification. Lines 475-480 discuss the impact of these mechanisms on precipitation structure, especially over the mountain. We have changed the language to clarify that we discuss this topic briefly in the paper.

**Comment 10:**

Line 589, this is not clear. Say from the start of this topic that you are referring to work does by Skamarock et al 2021. Otherwise you seemed to be saying things without evidence.

> **Response:** We have rearranged the paragraph beginning at line 591 to clarify which information is drawn from Skamarock et al. (2021).

**References**

Chen, C.-T. and Knutson, T.: On the Verification and Comparison of Extreme Rainfall Indices from Climate Models, J. Climate, 21, 1605–1621, https://doi.org/10.1175/2007JCLI1494.1, 2008.

Skamarock, W. C., Ong, H., and Klemp, J. B.: A Fully Compressible Nonhydrostatic Deep-Atmosphere Equations Solver for MPAS, Mon. Wea. Rev., 149, 571–583, https://doi.org/10.1175/MWR-D-20-0286.1, 2021.

Washington, W. M. and Baumhefner, D. P.: A method of removing Lamb waves from initial data for primitive equation models, J. Appl. Meteor., 14, 114–119, 1975.

70

**Response to Reviewer 2, Manuscript: egusphere-2023-376**

Owen Hughes and Christiane Jablonowski

**1    Author Comments**

We thank reviewer 2 for the highly appreciated constructive feedback, which improved the quality of the manuscript. When referring to line number edits made by the reviewer, we refer to the revised version of the manuscript. The key changes in the revised version of the manuscript are as follows:

- We have added a self-contained appendix defining Kessler physics. We have added a Fortran file to the code and data supplement, which allows Kessler to be added to new earth system models with minimal debugging.

- We have removed quantitative comparisons between the test case and atmospheric rivers.

- We revised the discussion of physics-dynamics coupling in Sect. 5.3 in the paper to tie it to the existing literature on the subject. We removed references to individual namelist settings. We think the resulting discussion is more generalizable beyond the specific models discussed in this paper.

- We have augmented the paper with several of the references suggested by the reviewer. These modifications help to relate our work to several bodies of existing literature.

**2    Response to Reviewer 2**

**Responses to Major Comments:**

**Comment M1:**

This manuscript proposes a new test case for atmospheric general circulation models for the evolution of baroclinic waves over topography. The proposed test case aims to examine the model performance of baroclinic waves over topographic mountains with or without moist processes. The description of the experimental design is presented except for cloud microphysics processes (see below). This paper is valuable for readers who can easily set up the test case by referring to this paper.

**Response:** We have added a new appendix containing the equations used to implement Kessler physics. However, due to the added difficulty of debugging a new Kessler implementation, we recommend using the `kessler.F90` file provided in our code supplement. We have explicitly stated this recommendation on lines 123 and 600 in the paper and have updated the code and data supplement.

**Comment M2:**

Although the description of the experimental design is clear, the purpose of the new test case is not clear enough, and the paper's purpose should be more focused. The experiments include the evolution of baroclinic waves, synoptic flow over mountains, and the moist effect on both. These are separately examined by the previous individual test cases. One new proposal might be the introduction of a ridge-type mountain. The authors may focus on the interaction of the flows over the ridges.

**Response:** The paper aims to introduce a new test technique for the dynamical cores of atmospheric General Circulation Models. The novelty of our approach is combining a baroclinic instability test case with idealized topographic barriers, which has been a missing link in the existing test case hierarchy. We now state this even more clearly at the end of the first paragraph of the introduction. A test case like this can be used for many purposes. From a 'dynamical core developer' perspective, the primary goal is to analyze the characteristics of the dynamical core's numerical schemes and its diffusion settings (e.g., via model intercomparisons), the use of the test as a debugging tool, and as a method

to shed light on physics-dynamics coupling issues in the moist configuration (e.g., are numerical oscillations created?). From a pure 'model user' perspective, a primary goal might be to analyze the impact of the topography on the flow field, which can be assessed both for dry or moist configurations. This has already been stated in the first paragraph of the introduction and the second paragraph of the conclusions in the original manuscript.

Our chosen examples provide snapshots for this broad application spectrum to illustrate how the new test case can be used. We can only pick a few examples, and they are mostly chosen from the 'model developer' perspective. Any future test case user is invited to focus their analyses on the interaction of the flow over mountains, as suggested by the reviewer. We indeed already started such an investigation (presented at a conference by Hughes and Jablonowski (2021)) where we varied the shapes and peak heights of the mountains to analyze the flow variations and also intercompared the evolution of the baroclinic waves with and without the mountains. However, such an in-depth flow interaction study is not within the scope of the current paper and warrants its own publication (in preparation). Please also see our response to S11, which shows an example figure (flow with and without the mountain) from the Hughes and Jablonowski (2021) presentation. We decided not to include such a figure here to not further lengthen the manuscript and widen the discussion.

**Comment M3:**

The moist process is an interesting aspect, but the inclusion of the moist process seems too much in this paper. The reviewer suggests the omission of the moist process from the paper.

**Response:** While the test case can be initialized in a dry atmosphere (see, e.g., Fig. 5 in the paper), the combination of a moist physics scheme and topographic forcing is part of what makes this test case novel. We have opted to keep the moist physics scheme in the specification of the test case.

**Comment M4:**

The proposed test case is conducted in a moderate resolution of AGCMs, and convective parameterization is generally used for moist convective processes. Most AGCMs generally do not have a rain category of the prognostic variable, and they cannot introduce the Kessler scheme. A more general and simpler choice for the moist process is the moist adjustment or the large-scale condensation with the saturation adjustment.

**Response:** We have also performed a model integration in the SE dycore with $0.5°$ grid spacing using large-scale condensation physics. Figures 2-4 show comparisons between model integrations where Kessler physics has been swapped for a large-scale condensation (LSC) scheme. Fig. 2 shows latitude-longitude profiles of temperature, large-scale precipitation, and surface pressure at day four. These fields show qualitative agreement, aside from increases in low-amplitude precipitation near the edges of precipitation bands in the LSC configuration. Fig. 3 shows MSLP and EKE for the two model configurations. The accumulation of EKE is decreased when LSC is used, but the rate of decrease of the minimum MSLP value is unchanged. Fig. 4 shows cross sections of temperature anomaly and vertical velocity at day 4. The vertical structure of the wave differs between LSC and Kessler. While there are differences between the two schemes, the differences are small enough that a test case with LSC would have similar characteristics to the one introduced in this paper. However, Kessler physics has been studied in a previous baroclinic wave test case (Ullrich et al., 2016). Because Kessler is not significantly more difficult to implement than LSC (see comment M5), having an equivalent physics scheme in both test cases is desirable. Therefore, we have created additional resources for introducing Kessler physics to a new model. But, if that fails, large-scale condensation could be used. However, Fig. 3(a) demonstrates that LSC-based integrations cannot be compared with the Kessler-based model integrations provided in the data supplement.

**Comment M5:**

If the authors nevertheless choose to introduce the Kessler scheme, the details of the scheme should be clearly defined in this manuscript. The original Kessler is shown in the cgs unit, and Klemp et al. (2015) do not clearly show the parameter values

except for the FORTRAN code. The authors should describe how to implement the Kessler scheme in AGCMs where a rain particle is not included as a prognostic variable.

    **Response:** For Ullrich et al. (2016), more than ten dynamical cores across multiple ESMs implemented Kessler physics to participate in a model intercomparison project. These models implemented Kessler during a short summer school with the help of a script containing a template Kessler implementation. To enable this kind of quick implementation for the test case in this paper, we have provided an updated template script containing an implementation of Kessler. We have added additional annotations in the code noting crucial implementation details that may differ from model to model, such as how vertical levels are ordered. We have also added a new appendix containing a detailed and self-contained description of the equations that define Kessler physics.

**Comment M6:**

Section 5.3 should be omitted. The readers are not interested in the model-specific namelist in the main text. This physics-dynamics argument does not seem to be a generic character. More examination must be added.

    **Response:** The authors believe that 5.3 is an important demonstration of a class of analyses that can be done with our test case. However, the focus on namelist arguments does make it difficult to extract general conclusions from this brief study of physics-dynamics coupling. We have updated the language to remove specific references to namelist parameters and focus instead on commonalities and differences between the dynamical cores. We have added a sentence contrasting the features that make SE an outlier to the strategies of the other dynamical cores on line 529.

**Comment M7:**

The four dynamical core results are presented in this paper. The results are just for comparison, and no physical insight is presented. Any test case is designed to know some specific characteristics of the models. The authors must extract and summarize the models' advantages and disadvantages. The test cases should generally be presented with standard analytic methods and a reference solution. These are not presented in this paper, and the readers cannot know how to evaluate their model results using the proposed test case.

    **Response:** Several moist test cases (Klemp et al., 2015; Ullrich et al., 2016; Thatcher and Jablonowski, 2016; Kurowski et al., 2015) have been introduced and used in the literature. Adding moist processes to these test cases precludes providing a reference solution. Therefore, evaluating how a dynamical core performs in these test cases is less straightforward than in the dry case. However, the aforementioned references provide examples of how comparisons can still be performed. Figure 4 and Fig. 11 in the paper show that our test case can be used to identify FV as an outlier despite the lack of a reference solution. Our data supplement provides all model integrations used to generate these figures. These datasets can be used to perform similar analyses for a new model which implements our test case.

    Constructing a hierarchy of model complexities requires the creation of test cases that stress-test physics dynamics coupling. The authors are unaware of existing literature that contains examples of constructing moist tests that "converge" with increasing resolution. Since our goal is to augment the intermediate-complexity part of the hierarchy, we feel that our analysis is on par with the analyses presented in, e.g., the above references relating to moist test cases.

**Responses to Specific Comments:**

**Comment S1:**

p. 2, L43-45: As a related reference for testing steep topography, the authors should refer to

  Satomura et al. (2003): Satomura, Takehiko & Iwasaki, Toshiki & Saito, Kazuo & Muroi, Chiashi & Tsuboki, Kazuhisa, 2003: Accuracy of Terrain Following Coordinates over Isolated Mountain: Steep Mountain Model Intercomparison Project (St-MIP). Annuals of Disaster Prevention Research Institute. 46. 337-346.

    **Response:** We thank the reviewer for the opportunity to augment the list of 2D nonlinear test case references with this work, which was added on line 47.

**Comment S2:**

p. 2, L44-46: The following literature is deserved the authors' notice in this context. Please refer to the following papers:
Yamazaki, H., and T. Satomura, 2010: Nonhydrostatic atmospheric modeling using a combined cartesian grid. Mon. Wea. Rev., 138, 3932–3945, .
Satoshi Masuda, and Keiichi Ishioka, 2015: A Method to Calculate Steady Lee-Wave Solutions with High-Accuracy, SOLA, 11, 85-89, .

> **Response: Yamazaki and Satomura (2010)**
> The emphasis on limited-area grids rather than global grids combined with the emphasis on how a specific vertical coordinate system is implemented put this reference outside of the scope of this paper. The references on line 47 identify a class of test cases that would be applicable for such a limited-area model, but our test case is constructed to test challenges that are particular to global grids. However, we acknowledge that both model types are necessary to simulate the dynamics of the atmosphere accurately. Additionally, a discussion of details for the vertical coordinate at the level given in the reference is not necessary for the analysis of the intercomparisons performed in this study. We have not identified a place where including this reference would add to the scientific background to this paper without dedicating significant time to explaining its conclusions. Therefore, we have elected not to include this reference.
>
> **Masuda and Ishioka (2010)**
> While methods such as these are remarkable for calculating 2D wave solutions on a Cartesian mesh, they do not generalize easily to 3D global solutions on a rotating earth. They also do not generalize to the case where moisture is present. Such a method is potentially valuable for constructing reference solutions for 2D topography on a Cartesian mesh. However, our literature review is primarily concerned with existing dynamical core test cases, especially those of intermediate complexity. We have not found a suitable place where this reference would naturally contribute to the scientific background of this study. Therefore, we have elected not to include the reference.

**Comment S3:**

p. 3, L68-70: An example of using the test case of Polvani et al. (2004), Iga et al. (2007) raised an issue that deserved discussion:
Iga, S., Tomita, H., Satoh, M., and Goto, K., 2007: Mountain-wave-like spurious waves associated with simulated cold fronts due to inconsistencies between horizontal and vertical resolutions. Mon. Weather Rev. 135, 2629–2641. doi:10.1175/MWR3423.1.

> **Response:** The discussion of applications of baroclinic life-cycle test cases has been augmented with reference to this example on line 78.

**Comment S4:**

p. 3, L90, "The mountains then act as triggers for baroclinic waves.": Baroclinic waves develop without mountains. The authors should clarify the roles of the mountains in this proposal. The initial development process of the baroclinic waves or the interaction between the baroclinic waves and the ridges can be analyzed separately.

> **Response:** The base state of this test case is designed so that, in the absence of topography, baroclinic waves will not be triggered until numerical errors induce enough deviation from the base state. This takes several days, and the triggered instabilities are not localized.

**Comment S5:**

p. 3, L79-86: A more fundamental test case of baroclinic waves can be traced back to the life cycle experiment by Brian Hoskins. The moisture process on the baroclinic waves can be tested using the baroclinic wave life cycle experiment.
Thorncroft, C.D., Hoskins, B.J. and McIntyre, M.E., 1993: Two paradigms of baroclinic-wave life- cycle behaviour. Q.J.R. Meteorol. Soc., 119: 17-55.

> **Response:** We have added a sentence on line 74 that references this paper, which mentions that a fundamental study of the life cycle of different baroclinic waves has been performed in other studies.

**Comment S6:**

p. 4, L116: The authors should clarify how the physical turbulence and numerical diffusions should be implemented. This information should be clarified for the examples used in this paper.

**Response:** The choice of which diffusion scheme to use is particular to the design and numerical characteristics of each dynamical core. While a test case such as ours can be used to select an appropriate set of such mechanisms, the implementation of individual mechanisms (of which there are many) is beyond the scope of the paper. A review of such mechanisms can be found in Jablonowski and Williamson (2011), but spans well over one hundred pages.

**Comment S7:**

p. 4, L124, "atmospheric river": Testing atmospheric rivers (AR) in AGCMs is a good point of test cases. However, the proposed test case is not ideal for testing AR. The mountain is an additional factor for testing AR. The simpler baroclinic wave test case without a mountain is suitable for testing AR.

**Response:** The reviewer makes a valuable point that AR are complex phenomena, which deserve a test case of their own. Specialized, detailed analyses which are tailored to the computational difficulties of simulating atmospheric rivers would be necessary to do this topic justice. We have removed quantitative comparisons to statistics from Hagos et al. (2015) (i.e. changes to lines 236 and 407 ). However, we leave in two qualitative comparisons of our test case to topographic forcing of atmospheric rivers. We believe that these comparisons help link the extreme precipitation and topography that make our test case difficult to simulate to real-world phenomena which face qualitatively similar difficulties when they are represented in simulations.

**Comment S8:**

p. 6, L139: The authors can estimate the growth rate of baroclinic instability.

**Response:** While a linearized analysis of the growth rate of the baroclinic instability would be of dynamical interest, the inclusion of moist physics makes this analysis minimally relevant to the development of the wave after precipitation develops. This may be valuable for specific analysis of the dynamics in the companion paper on topographic forcing. However, the authors feel that the length of additional explanation necessary to derive and contextualize this quantity (see e.g. Ullrich et al. (2015)) is not justified since it is minimally relevant to how this test case is used.

**Comment S9:**

p. 12, L316, "with nominal grid spacings of 1 deg (ne30)": It is confusing. If the ne30 setting has a 30×30 supporting elements per cubed-sphere face, the grid spacing is 3 deg. The authors should clarify the grid spacing.

**Response:** The SE dynamical core is based on a continuous Galerkin spectral finite element method. The horizontal order for SE in CAM is chosen to have four Gauss-Lobatto-Legendre (GLL) points on each side of a quadrilateral element. Solutions given by the SE method are computed at GLL points. On a quasi-uniform cubed sphere grid, the nominal grid spacing between GLL points is divided by three, which gives the $1°$ nominal grid spacing. Since these details are not immediately relevant to the analysis in this study, line 325 has been rewritten in more general terms with a reference to Lauritzen et al. (2018), which contains a comprehensive treatment of SE resolution.

**Comment S10:**

p. 13, Figure 4: The label "(g)" within the figures appears twice. The second one should be "(h)".

**Response:** We thank the reviewer for the astute observation. The figure label has been corrected.

**Comment S11:**

p. 15, L365-368: The authors can conduct experiments by changing $h = 0$ (including $h_0 = 0$) to examine the effect of the topography.

   **Response:** While the authors agree that careful dynamical study of the flow in the absence of topography is valuable, it is beyond the scope of the current paper. A sample figure drawn from a companion paper that is in preparation on the dynamical impacts of the mountain is shown in Fig. 1. We intend to study the impact of topographic profile on growth rate and baroclinic life cycle in this companion paper.

**Comment S12:**

p. 21, L468-469, "The most obvious difference is that the MPAS high pressure systems with MSLP values over 1010 hPa occupy visibly larger areas.": This result is interesting. Does this high- pressure anomaly emerge in a dry model?

   **Response:** We have included Fig. 5 in this document to compare MPAS model integrations in a moist and a dry atmosphere. The high pressure regions in each MPAS configuration are visibly larger than their SE counterparts. This indicates that these differences may result from how MPAS treats dynamics and is separate from physics-dynamics coupling issues. Further investigation of the dry wave may show why this difference is observed. However, that is beyond the scope of the current paper.

**Comment S13:**

p. 21, L471, "the Kessler physis": If the Kessler scheme is proposed to be used as a test case, the comprehensive analysis method should be presented. The distributions of rain and cloud mixing ratios and the tendency terms of the conversion processes are suggested. The evaluations of cloud cover and the cold pool are interesting aspects of model comparison.

   **Response:** While one goal of this publication is to provide examples of ways that this test case can be used to compare models, we do not claim that our analyses are the only intercomparisons that can be done. An analysis of tendency terms of the conversion processes would be scientifically valuable but is beyond the scope of the current paper. In particular, conversion between moisture species at a given time step is primarily handled within the physics parameterization itself. These tendencies, therefore, do not depend strongly on the dynamical core, which is what our test case is designed to examine.

   However, Fig. 10 shows an intercomparison using the cloud liquid mixing ratio. In Kessler physics, this is the closest analog to cloud cover. Indeed, it is interesting that this evaluation shows some of the most striking differences between dynamical cores. However, a comprehensive analysis of these differences would be prohibitively long due to the complexity of how dynamics and physics interact in each dynamical core. The cold pool is a phenomenon that is most relevant to simulations of storms at convection-resolving scales. While there may be analogs in moderate-resolution models, it is not a common point of comparison in other test cases that use Kessler physics.

**Comment S14:**

p. 23, Section 5.3 The reviewer suggests the omission of this subsection. The model-specific namelist variable should be avoided in the main text.

   **Response:** See response to Comment M6

**Comment S15:**

p. 23, L517: What is the definition of "the numerical ringing"?

   **Response:** This section has been updated to remove references to "numerical ringing" and consistently describe the phenomenon as "grid-scale oscillations." We related it to Gibbs Ringing several lines after it is referenced here, on line 524.

**Comment S16:**

p. 27, L614-615: What is the physical meaning of $\tau_1$ and $\tau_2$? Any reference for this specification?

    **Response:** The derivation of $\tau_1$ and $\tau_2$ can be found in Staniforth and White (2011). Because these definitions have been included for reproducibility, we opted to have a reference to this paper. The interested reader can consult this reference for a closer analysis of how they construct a family of temperature structures. We have added a sentence to the appendix directing readers to this reference.

245

**References**

Hagos, S., Leung, L. R., Yang, Q., Zhao, C., and Lu, J.: Resolution and Dynamical Core Dependence of Atmospheric River Frequency in Global Model Simulations, J. Climate, 28, 2764–2776, 2015.

250 Jablonowski, C. and Williamson, D. L.: The Pros and Cons of Diffusion, Filters and Fixers in Atmospheric General Circulation Models, in: Numerical Techniques for Global Atmospheric Models, edited by Lauritzen, P. H., Jablonowski, C., Taylor, M. A., and Nair, R. D., vol. 80 of *Lecture Notes in Computational Science and Engineering*, pp. 381–493, Springer, 2011.

Klemp, J. B., Skamarock, W. C., and Park, S.-H.: Idealized global nonhydrostatic atmospheric test cases on a reduced-radius sphere, J. Adv. Model. Earth Syst., 7, 1155–1177, https://doi.org/10.1002/2015MS000435, 2015.

255 Kurowski, M. J., Grabowski, W. W., and Smolarkiewicz, P. K.: Anelastic and Compressible Simulation of Moist Dynamics at Planetary Scales, J. Atmos. Sci., 72, 3975–3995, 2015.

Lauritzen, P. H., Nair, R. D., Herrington, A. R., Callaghan, P., Goldhaber, S., Dennis, J. M., Bacmeister, J. T., Eaton, B. E., Zarzycki, C. M., Taylor, M. A., Ullrich, P. A., Dubos, T., Gettelman, A., Neale, R. B., Dobbins, B., Reed, K. A., Hannay, C., Medeiros, B., Benedict, J. J., and Tribbia, J. J.: NCAR Release of CAM-SE in CESM2.0: A Reformulation of the Spectral Element Dynamical Core in Dry-
260 Mass Vertical Coordinates With Comprehensive Treatment of Condensates and Energy, J. Adv. Model. Earth Syst., 10, 1537–1570, https://doi.org/10.1029/2017MS001257, 2018.

Staniforth, A. and White, A. A.: Further non-separable exact solutions of the deep- and shallow-atmosphere equations, Atmospheric Science Letters, 12, 356–361, https://doi.org/10.1002/asl.349, 2011.

Thatcher, D. R. and Jablonowski, C.: A moist aquaplanet variant of the Held-Suarez test for atmospheric model dynamical cores, Geoscientific
265 Model Development, 9, 1263–1292, https://doi.org/10.5194/gmd-9-1263-2016, 2016.

Ullrich, P. A., Jablonowski, C., Reed, K., Zarzycki, C., Lauritzen, P., Nair, R., Kent, J., and Verlet-Banide, A.: Dynamical Core Model Intercomparison Project (DCMIP2016) Test Case Document, Tech. rep., University of California, Davis, available at http://www-personal. umich.edu/~cjablono/dycore_test_suite.html and https://github.com/ClimateGlobalChange/DCMIP2016, 2016.

**FV3C192L30_nord_2_hord_5, 2000 m ridge**

[Figure]

**Figure 1.** Top row shows Latitude-longitude cross sections of surface pressure comparing (a) a simulation containing two mountains to (b) a simulation without a second mountain. Panel (c) shows the difference between (a) and (b).

[Figure]

**Figure 2.** Latitude-longitude cross-sections of the baroclinic waves in the SE dynamical core on a $0.5°$ degree grid at day 5. The left column shows model runs that use Large Scale Condensation physics. The right column shows model runs as in the manuscript that use Kessler physics. Top row: mean sea level pressure, middle row: precipitation rate, bottom row: 850 hPa temperature. The contour lines indicate the location of the mountain ridges.

[Figure]

**Figure 3.** Time series of baroclinic wave summary statistics from the moist SE model with nominal $0.5°$ (ne60) grid spacing using Kessler physics or a Large Scale Condensation scheme. (a) Eddy Kinetic Energy, and (b) point-wise minimum MSLP, which is a proxy for the amplification of the baroclinic wave.

[Figure]

**Figure 4.** Longitude-height cross-sections at $45°$ N of the (top) temperature perturbation and (bottom) vertical pressure velocity $\omega$ for the (a,c) Large Scale Condensation and (b,d) Kessler model simulations. SEne60 (50 km) model integrations at day 4 are shown.

[Figure]

**Figure 5.** Latitude-longitude cross sections of MSLP for SE and MPAS are shown. The left column contains SE, and the right column contains MPAS simulations. The moist simulations are in the top row, and the dry simulations are in the bottom row.